# Insights into stem Batomorphii: A new holomorphic ray (Chondrichthyes, Elasmobranchii) from the upper Jurassic of Germany

Julia Türtscher[1,2]*, Patrick L. Jambura[1,2], Frederik Spindler[3], Jürgen Kriwet[1,2]

**1** Department of Palaeontology, Faculty of Earth Sciences, Geography and Astronomy, Evolutionary Research Group, University of Vienna, Vienna, Austria, **2** Vienna Doctoral School of Ecology and Evolution (VDSEE), University of Vienna, Vienna, Austria, **3** PALAEONAVIX, Kipfenberg, Germany

\* tuertscher.julia@gmail.com

**Data Availability Statement:** All relevant data are within the manuscript and its Supporting Information files.

## Abstract

The Late Jurassic fossil deposits of southern Germany, collectively known as the 'Solnhofen Archipelago', are one of the world's most important sources of Mesozoic vertebrates. Complete skeletons of cartilaginous fishes (Chondrichthyes), whose skeletal remains are rare in the fossil record and therefore all the more valuable, are represented, among others, by exceptionally well-preserved rays (superorder Batomorphii). Despite their potential for research in several areas, including taxonomy, morphology, ecology, and phylogeny, the number of studies on these chondrichthyans is still very limited. Here, we identify a previously unknown ray, †*Apolithabatis seioma* gen. et sp. nov., which represents the first record of a ray species from the upper Kimmeridgian of Painten, Germany, and thus the oldest Late Jurassic ray taxon from Germany based on skeletal remains. This new batomorph is characterised by a unique body shape and a combination of skeletal features that distinguish it readily from all other known Late Jurassic rays. Two different morphometric approaches confirm differences in body shape and proportions to all known Late Jurassic conspecifics. We thus extend the recent taxonomic revision of these rays and include all described holomorphic specimens in a phylogenetic framework using strict cladistic principles. The phylogenetic analysis reveals all Late Jurassic batomorphs to represent a monophyletic group, for which we introduce the new order Apolithabatiformes, which is sister to all other batomorphs representing a stem group. While the phylogenetic relationships within Apolithabatiformes ord. nov. remain largely unresolved, †*Apolithabatis* gen. nov. is placed as the sister to †*Aellopobatis*. This highlights that, despite considerable progress in our understanding of the diversity and phylogeny of early rays, difficulties remain in establishing robust relationships within batomorphs. We therefore emphasise the importance of comprehensive studies of completely preserved fossil cartilaginous fishes to obtain a better understanding of chondrichthyan evolution and their systematics in deep time.

**Funding:** This research was funded in whole by the Austrian Science Fund (FWF) P33820 and P35357. For the purpose of open access, the authors have applied a CC BY public copyright licence to any Author Accepted Manuscript version arising from this submission. The funders had no role in study design, data collection and analysis, decision to publish, or preparation of the manuscript. https://www.fwf.ac.at/en/.

**Competing interests:** The authors have declared that no competing interests exist.

## Introduction

Chondrichthyans or cartilaginous fishes, are among the most abundant remains in the vertebrate fossil record but are usually represented only by isolated teeth (Hubbel, 1996 [1]; Maisey, 2012 [2]). Complete specimens, conversely, are very rare due to the poor preservation potential of their cartilaginous endoskeletons. Such rare 'holomorphic' fossils provide insights into body shapes, skeletal anatomy, and even soft tissue composition of long extinct taxa, allowing researchers to explore a greater variety of ecological and morphological traits in deep time. The oldest currently known holomorphic ray fossils are from the Jurassic period (Maisey et al., 2020 [3]; Türtscher et al., 2024 [4]), which evidently was an important time interval in shark and ray evolution (Guinot and Cavin, 2016 [5]). Yet, our knowledge about diversity and disparity of Jurassic rays, as well as about their systematic placement within the chondrichthyan phylogeny, is still very limited (Underwood, 2006 [6]).

The scarcity of well-preserved holomorphic chondrichthyans makes deposits that yield such extraordinary fossils (so-called 'Konservat-Lagerstätten') particularly valuable sources of information about early sharks and rays. Among the best known of these deposits is the Late Jurassic 'Solnhofen Archipelago' located in southern Germany, which consists of several individual sites of Kimmeridgian-Tithonian age (Villalobos-Segura et al., 2023 [7]). The diverse vertebrate fauna of these extraordinary fossil sites contains several chondrichthyan taxa, including at least three holocephalian, three hybodontiform, fifteen selachimorph, and two batomorph genera (Villalobos-Segura et al., 2023 [7]). Until recently, it was assumed that two ray genera occurred in the 'Solnhofen Archipelago', i.e., †*Asterodermus* and †*Spathobatis* (Kriwet and Klug, 2015 [8]; Villalobos-Segura et al., 2023 [7]). However, a recent comprehensive study of the Late Jurassic batomorphs of Europe revealed that none of the specimens from the German deposits belong to †*Spathobatis*, which thus appears to be restricted to France. Instead, what had previously been described as a large morphotype of †*Spathobatis* was found to represent a different taxon, †*Aellopobatis* (Türtscher et al., 2024 [4]). Skeletal remains of both †*Asterodermus* and †*Aellopobatis* are known from several of the Tithonian sites in southern Germany (Türtscher et al., 2024 [4]), but to our knowledge none have been described from any of the Kimmeridgian deposits up to now. It should be noted, however, that precise provenance information is not always available for fossils from the 'Solnhofen Archipelago', especially for those collected decades ago (Villalobos-Segura et al., 2023 [7]).

Here, we describe the first holomorphic ray from the locality of Painten within the 'Solnhofen Archipelago', where upper Kimmeridgian to lower Tithonian strata are exposed and yield excellently preserved fossils. The specimen described here represents a hitherto unknown taxon from upper Kimmeridgian strata. We analysed the specimen employing morphometric approaches as proposed by Türtscher et al. (2024) [4] for taxonomic identification and included it into a phylogenetic analysis based on a revised character matrix that includes all described Late Jurassic ray species known from holomorphic specimens to resolve its phylogenetic interrelationships.

## Geological setting

The specimen described here was recovered from a quarry of the Rygol Company near the village of Painten, which is located in the northern part of the so-called 'Paintener Wanne', a basin within the 'Solnhofen Archipelago' extending over an area of about 15 x 12 km (Albersdörfer & Häckel, 2015 [9]). This basin is located in the southeastern part of the Franconian Alb in central Bavaria, which forms, together with the Swabian Alb in the West, a low mountain range consisting predominantly of Early to Late Jurassic marine sedimentary deposits. Painten is, together with Daiting, Mörnsheim, Solnhofen, Eichstätt, and Schamhaupten in the west as

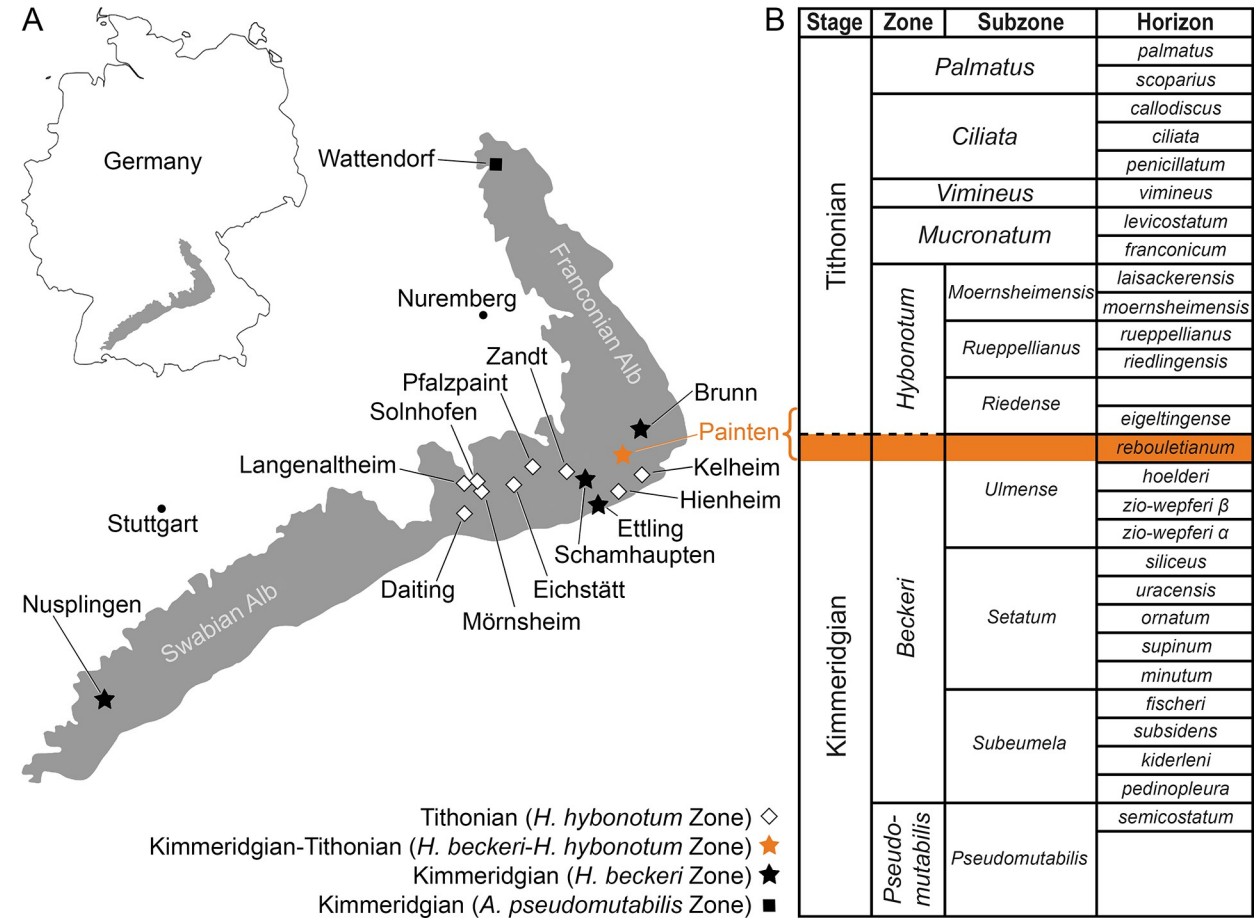

**Fig 1. Geographical setting and stratigraphy of Painten.** A) Geographical map of the 'Solnhofen Archipelago' and Nusplingen, modified from Villalobos-Segura et al. (2023) [7] under CC BY 4.0; outline of Germany created with the R package *maps* (Becker et al., 2018) [11]; for a palaeogeographical map with biostratigraphical information on the 'Solnhofen Archipelago' see Villalobos-Segura et al. (2023) [7]. B) Stratigraphic section of the Upper Jurassic (upper Kimmeridgian to lower Tithonian) sediments of the 'Solnhofen Archipelago' (southern Germany), the sequence exposed at Painten is indicated by a bracket. Note that the new batomorph fossil is from the *rebouletianum*-horizon within the *Lithacoceras ulmense* Subzone of the Kimmeridgian (highlighted).

well as Kelheim, Jachenhausen, Zandt, and Brunn in the south and northeast, respectively, one of the best-known fossiliferous sites in the Franconian Alb (Fig 1A). All deposits are generally composed of fine-grained limestones representing the so-called 'Plattenkalks', but local sedimentological differences occur (Viohl, 2015 [10]). Moreover, the various 'Plattenkalks' are not of the same age, but belong to at least four distinct formations: Torleite (Malm Epsilon), Geisental (Malm Zeta 1), Painten (Malm Zeta 1), Altmühltal (Malm Zeta 2) and Mörnsheim (Malm Zeta 3) formations, which range in age from the upper Kimmeridgian to lower Tithonian.

In the fossiliferous quarry of the Rygol Company near the village of Painten, a 5.9m thick layer of the renowned 'Kieselplattenkalk' is exposed. This layer consists of laminated, fine-grained, silicified limestone, intercalated with graded turbidite horizons composed of carbonate debris. Stratigraphically, these very dense and compact sediments are placed into the Arnstorf Member of the Torleite Formation, which has been assigned to the *Hybnoticeras beckeri* ammonite Zone and *Lithacoceras ulmense* Subzone (Fig 1B), indicating a latest Kimmeridgian age (Albersdörfer & Häckel, 2015 [9]).

The specimen that forms the focus of the present study was recovered in very hard layers close to the base of the exposed section of the 'Kieselplattenkalk'. Traditional excavation using tools, therefore, was not possible, and a small quantity of explosives was detonated six metres below the surface, creating cracks in the rock without disorganising the layers. However, the top layer of rock was thrown into the air by the explosion and broke into several slabs. One of these slabs contained the specimen described here.

## Material and methods

### Material

A total of 55 specimens of articulated Late Jurassic batomorphs were used in the present study (S1 Table in S2 File). The age of the specimens ranges from the upper Kimmeridgian (Cerin, France; 'Solnhofen Archipelago' [Painten], Germany) to the lower Tithonian ('Solnhofen Archipelago' [Eichstätt, Kelheim, Solnhofen, Zandt], Germany; Kimmeridge, UK), and represent all hitherto known batomorph morphotypes of these fossil sites (see Türtscher et al., 2024 [4]). Not available for this study, however, were †*Spathobatis*? *morinicus* Sauvage, 1873 [12] from the lower Tithonian of Boulogne-sur-Mer, France, a holomorphic batomorph based on a single specimen (Sauvage, 1873 [12]; Cavin et al., 1995 [13]), as well as an unnamed batomorph from the middle Tithonian of Argentina (Cione, 1999 [14]). Published figures of these two specimens were not considered to be of high enough quality to be used for measurements.

All specimens were photographed with a digital camera positioned orthogonally to each specimen to avoid doubtful results due to a misaligned angle. Some specimens were examined under ultraviolet light following the technique described in Tischlinger & Arratia (2013) [15] for better identification of specific skeletal structures. Daggers before taxon names indicate extinct taxa.

**Institutional abbreviations.** **AMNH**, American Museum of Natural History, New York, USA; **BMMS**, Bürgermeister-Müller-Museum, Solnhofen, Germany; **CM**, Carnegie Museum of Natural History, Pittsburgh, USA; **DMA**, Dinosaurier Museum Altmühltal, Denkendorf, Germany; **HMNS**, Houston Museum of Natural Science, Houston, Texas, USA; **JME**, Jura-Museum Eichstätt, Eichstätt, Germany; **LF**, Lauer Foundation for Paleontology, Science and Education, Wheaton, Illinois, USA; **MB**, Museum Bergér, Eichstätt, Germany; **MCZ**, Museum of Comparative Zoology, Cambridge, Massachusetts, USA; **MGL**, Musée cantonal de Géologie, Lausanne, Switzerland; **MDC**, Musée des Confluences, Lyon, France; **MJML**, Museum of Jurassic Marine Life, Kimmeridge, UK; **MNB**, Museum für Naturkunde, Berlin, Germany; **MNHN**, Muséum National d'Histoire Naturelle, Paris, France; **NHMUK**, Natural History Museum, London, UK; **NRM**, Naturhistoriska Riksmuseet, Stockholm, Sweden; **RBINS**, Royal Belgian Institute of Natural Sciences, Brussels, Belgium; **SMNK**, Staatliches Museum für Naturkunde Karlsruhe, Karlsruhe, Germany; **SNSB-BSPG**, Bayerische Staatssammlung für Paläontologie und Geologie, Munich, Germany; **TM**, Teylers Museum, Haarlem, Netherlands.

**Anatomical abbreviations.** **ac**, antorbital cartilage; **bp**, basipterygium; **br**, branchial arches; **c**, vertebral centra; **cf**, caudal fin; **cr**, compound radial; **d1**, first dorsal fin; **d2**, second dorsal fin; **hs**, haemal spine; **mk**, Meckel's cartilage, **ms**, mesopterygium; **mt**, metapterygium; **nc**, nasal capsule; **ns**, neural spine; **pb**, puboischiadic bar; **pp**, propterygium; **pq**, palatoquadrate; **r**, ribs; **ra**, pectoral fin radials; **rap**, pelvic fin radials; **ro**, rostrum; **sc**, scapulocoracoid; **syn**, synarcual.

**Nomenclatural acts.** The electronic edition of this article conforms to the requirements of the amended International Code of Zoological Nomenclature, and hence the new names contained herein are available under that Code from the electronic edition of this article. This published work and the nomenclatural acts it contains have been registered in ZooBank, the

online registration system for the ICZN. The ZooBank LSIDs (Life Science Identifiers) can be resolved and the associated information viewed through any standard web browser by appending the LSID to the prefix ""http://zoobank.org/"". The LSID for this publication is: urn:lsid: zoobank.org:pub:B2CEE8F7-4F41-459D-A45B-8C6AB2024F24. The electronic edition of this work was published in a journal with an ISSN, and has been archived and is available from the following digital repositories: PubMed Central, LOCKSS.

## Linear measurements

As basis of our traditional morphometrics analyses, we used the dataset of Türtscher et al. (2024) [4], who investigated 30 holomorphic specimens representing †*Aellopobatis bavarica*, †*Asterodermus platypterus*, †*Belemnobatis sismondae*, and †*Spathobatis bugesiacus*. As shown by Türtscher et al. (2024) [4], †*Belemnobatis sismondae* can be clearly distinguished from all other taxa both morphologically and morphometrically. The specimen examined here can be clearly distinguished from †*B. sismondae* by its qualitative characters, and we therefore excluded the latter species from our morphometric analyses, resulting in the removal of six specimens belonging to †*B. sismondae* from the original dataset, but added the holotype of the newly described species (DMA-JP-2010/ 007), resulting in a new dataset containing 25 specimens (S1 Table in S2 File).

We used ImageJ (version 1.53t) to take 27 different measurements (see Fig 2) on each specimen based on photographs to the nearest 0.001 mm. Each measurement was taken three times and the mean values were calculated to reduce measurement errors. Most images included a scale bar with 1 mm increments; for the images without a scale bar, pixels were measured because the subsequent analyses used relative, rather than metric, measurements. The measurements taken were adjusted as percentages of the disc width (% DW) of each individual for traditional morphometric analysis (see below) as well as for species comparison (see 'Systematic Palaeontology' section below). In addition, a second traditional morphometric analysis was performed, with measurements adjusted as percentages of the disc length (% DL) of each individual (see S4 File). Instead of using the percentage of the total length for species comparison, we decided to use the percentages of the disc width and disc length for two reasons: (1) several specimens are not completely preserved, so total length could not be measured, and (2) completely preserved specimens often do not lie straight but are bent, and measuring the fossil with the tail not straightened can lead to measurement errors. However, the measurements taken from specimen DMA-JP-2010/007 were additionally adjusted as percentages of the total length (% TL) for the species description (see Systematic Palaeontology section below).

It was not possible to obtain all measurements from some of the specimens included, e.g., due to incomplete preservation, resulting in our dataset also containing missing data. We first divided the dataset into *a priori* sorted subsets based on the initial qualitative classification of specimens to reduce the noise caused by missing data in subsequent analyses and imputed the missing values of each subset using a regularized iterative PCA algorithm with the function *imputePCA* in the R package *missMDA* (Josse & Husson, 2016 [16]). The subsets with the implemented data were merged into a dataset containing all specimens and subjected to a Principal Component Analysis (PCA).

Several statistical tests were carried out to explore differences between the groups. First, a Shapiro-Wilk test for normal distribution was applied to each measurement. Subsequently, we created subsets including either normally distributed or non-normally distributed relative measurements, respectively. The dataset containing non-normally distributed measurements was further examined with non-parametric tests. First, a Kruskal-Wallis test was applied to assess the differences of each measurement among groups. A pairwise Wilcoxon rank sum test with Bonferroni corrections for multiple comparisons for the differences between groups was

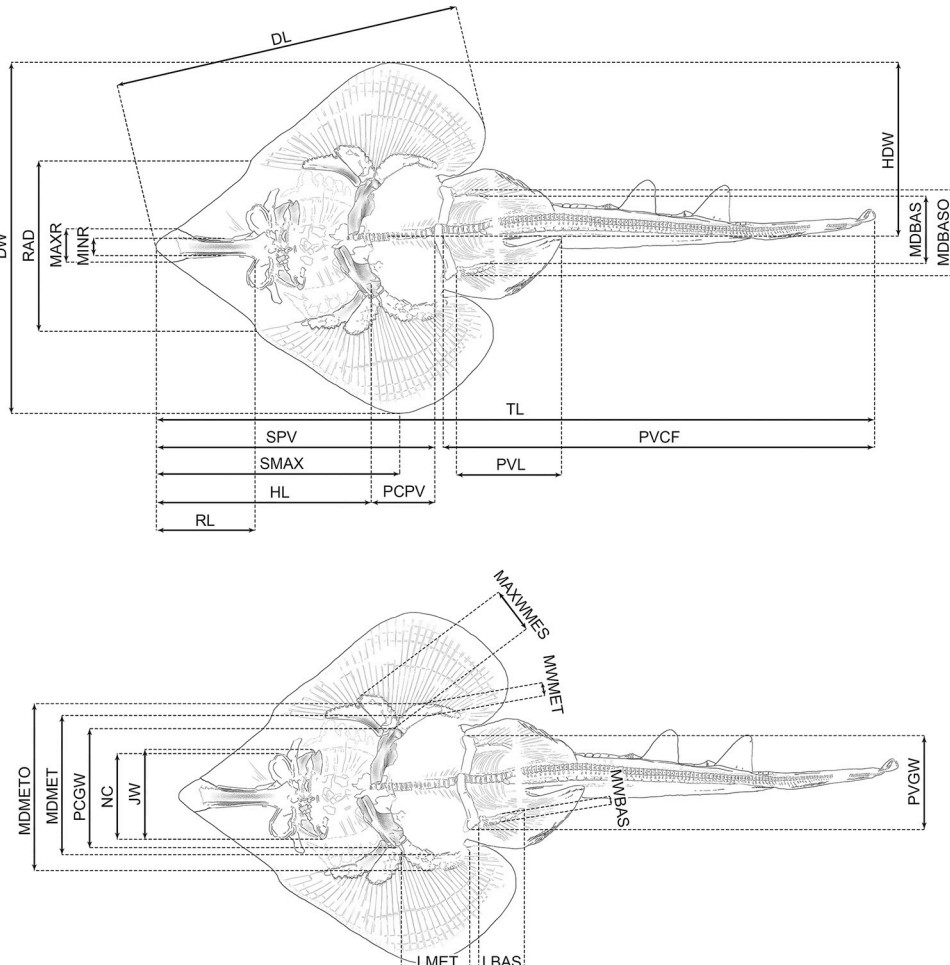

**Fig 2. Illustrations of DMA-JP-2010/007 showing the measurements used for traditional morphometric analyses.**
Note that the illustrations are mirror images of the original fossil. Abbreviations: DL, disc length; DW, disc width; HDW, half disc width; HL, head length; JW, jaw width; LBAS, length of basipterygia; LMET, length of metapterygia; MAXR, maximum rostrum width; MAXWMES, maximum width of mesopterygia; MDBAS, inner maximum distance between basipterygia; MDBASO, outer maximum distance between basipterygia; MDMET, inner maximum distance between metapterygia; MDMETO, outer maximum distance between metapterygia; MINR, minimum rostrum width; MWBAS, maximum width of basipterygia; MWMET, maximum width of metapterygia; NC, nasal capsules maximum width; PCGW, pectoral girdle width; PCPV, pectoral girdle to pelvic girdle; PVCF, pelvic girdle to caudal fin tip; PVGW, pelvic girdle width; PVL, pelvic fin length; RAD, span between anteriormost fin radials; RL, rostrum length; SMAX, distance from the tip of the snout to the point of maximum disc width; SPV, snout to pelvic girdle; TL, total length.

conducted subsequently. The normally distributed measurements were examined for differences with an Analysis of Variance (ANOVA) and a subsequent pairwise comparison.

The R code published by Türtscher et al. (2024) [4] was used for all analyses, which were performed using R 3.6.3 (R Core Team, 2013 [17]) and RStudio 1.2.5019 (R Studio Team, 2019 [18]). Related data is available in S10 File. Plots were created using the R packages *ggplot2* (Wickham, 2016 [19]) and *viridisLite* (Garnier, 2018 [20]).

## Geometric morphometrics

**Head outline.** As basis of our analyses, we used the dataset of Türtscher et al. (2024) [4], who investigated 21 holomorphic specimens containing the species †*Aellopobatis bavarica*,

†*Asterodermus platypterus*, †*Belemnobatis sismondae*, †*Kimmerobatis etchesi*, and †*Spathobatis bugesiacus*. †*Belemnobatis sismondae*, which only occurs in the Kimmeridgian of Cerin (France) was excluded from the present study (see above), but we added two new specimens of †*Aellopobatis bavarica*, one new specimen of †*Asterodermus platypterus*, one new specimen of †*Spathobatis bugesiacus*, and the only specimen of the new taxon to the data matrix (see S1 Table in S2 File).

In total, we performed 2D landmark-based geometric morphometrics on 22 specimens (S1 Table in S2 File). Five homologous landmarks were digitized using the software tpsDIG2 (v. 2.31; Rohlf, 2017 [21]). Landmarks (1) and (4) are located at the point where the extension of the first radial of the propterygium reaches the edge of the disc, on the right and left pectoral fin respectively; (2) and (3) are located at the notches that indicate the connection between the base of the rostrum and the nasal capsules; (5) is located at the tip of the rostrum. Additionally, 36 semilandmarks were digitized between the homologous landmarks to capture the overall shape of the cranial region. They are arranged in two curves of 18 points each, one between landmarks (1) and (5) and one between landmarks (5) and (4). These two curves describe the shape of the head from the tip of the snout to the first pectoral radial (Fig 3A). A Generalized Procrustes Analysis (GPA) was performed on the landmark coordinates to minimize the variance caused by factors such as size, orientation, location, and rotation (Rohlf & Slice, 1990 [22]). For minimization of the bending energy, the semilandmarks were allowed to slide (Bookstein, 1997 [23]). Subsequently, the aligned coordinates were subjected to a Principal Component Analysis (PCA) to assess the shape variation of the snouts of the specimens. Shape and size differences between the groups were estimated with a Procrustes ANOVA with 10.000 permutations, followed by pairwise comparisons between the groups, with the functions *procD.lm* and *pairwise* considering the distances between means in the R packages *geomorph* (v. 4.0.4; Adams et al., 2016 [24]) and *RRPP* (Collyer & Adams, 2018 [25]).

The R code published by Türtscher et al. (2024) [4] was used for all analyses, which were performed using R 3.6.3 (R Core Team, 2013 [17]) and RStudio 1.2.5019 (R Studio Team, 2019

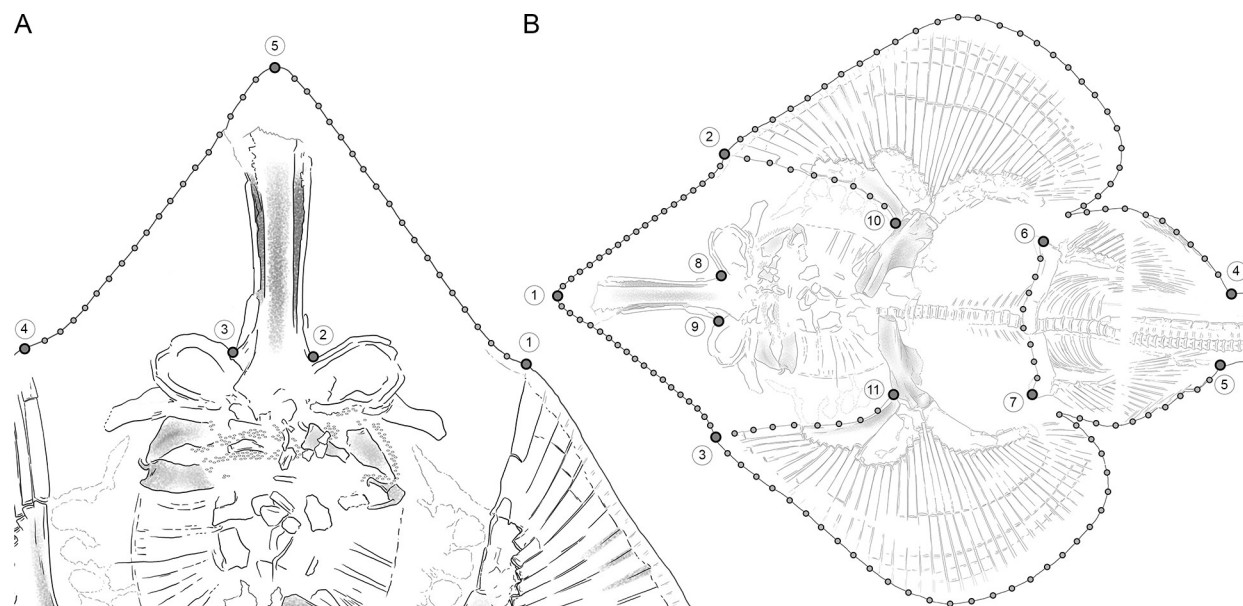

**Fig 3. Location of the true landmarks (large dots in dark grey) and semilandmarks (small dots in light grey) for the two geometric morphometric analyses.** A) Head-outline. B) Complete body.

[18]). Related data is available in S10 File. Plots were created using the R packages *ggplot2* (Wickham, 2016 [19]) and *viridisLite* (Garnier, 2018 [20]).

**Complete body.**   This analysis included 10 specimens that met the following criteria: very good preservation of the body outline from the tip of the snout to the posterior end of the pelvic fins, the pelvic girdle, the point of attachment of the rostral cartilage to the neurocranium, and the internal boundaries of the pectoral fins from the anterior end to the point of articulation of the propterygium with the scapulocoracoid (S1 Table in S2 File). Eleven homologous landmarks were digitized using the software tpsDIG2 (v. 2.31; Rohlf, 2017 [21]). Landmark (1) is located at the tip of the snout; (2) and (3) are located at the point where the extension of the first radial of the propterygium reaches the edge of the disc, on the right and left pectoral fin respectively; (4) and (5) are located at the point where the pelvic fins join the trunk; (6) and (7) are located at the anterior lateral-most points of the pelvic girdle; (8) and (9) are located at the notches that indicate the connection between the base of the rostrum and the nasal capsules; (10) and (11) are located at each connecting point from the propterygia to the scapulocoracoid. Additionally, 136 semilandmarks were digitized between the homologous landmarks to capture the overall shape of the body outline, the pelvic girdle, and the anterior internal pectoral fin boundaries. They are arranged in seven curves; 18 semilandmarks each are located between landmarks (1) and (2) as well as between (1) and (3); 38 semilandmarks each are located between landmarks (2) and (4) as well as between (3) and (5); these curves describe the body outline from the tip of the snout to the posterior end of the pelvic fins; eight semilandmarks are located between landmarks (6) and (7), describing the anterior outline of the pelvic girdle; eight semilandmarks each are located between landmarks (2) and (10) as well as between (3) and (11) (Fig 3B). A GPA was performed on the landmark coordinates, and the semilandmarks were allowed to slide. The aligned coordinates were then subjected to a PCA. Shape and size differences between the groups were estimated with a Procrustes ANOVA with 10.000 permutations, followed by pairwise comparisons between the groups, with the functions *procD.lm* and *pairwise* considering the distances between means in the R packages *geomorph* (v. 4.0.4; Adams et al., 2016 [24]) and *RRPP* (Collyer & Adams, 2018 [25]).

The R code published by Türtscher et al. (2024) [4] was used for all analyses, which were performed using R 3.6.3 (R Core Team, 2013 [17]) and RStudio 1.2.5019 (R Studio Team, 2019 [18]). Related data is available in S10 File. Plots were created using the R packages *ggplot2* (Wickham, 2016 [19]) and *viridisLite* (Garnier, 2018 [20]).

**Remarks.**   It should be emphasized that statistical analyses involving groups consisting of a single specimen only serve to support the results and interpretations and must be taken with caution due to the limited sample size. In order to obtain robust statistical results, ideally several specimens of a group should be analyzed, which is unfortunately not possible at this time for the two Late Jurassic batomorph taxa †*Apolithabatis seioma* gen. et sp. nov. and †*Kimmerobatis etchesi* due to the small number of known specimens.

## Phylogenetic analyses

For phylogenetic analyses, we adopted and modified the character matrix of Villalobos-Segura et al. (2022) [26]. To identify potential rogue taxa in the original dataset, an initial phylogenetic analysis was conducted on the unmodified data set of Villalobos-Segura et al. (2022) [26] in TNT 1.6 on MacOS (Goloboff et al., 2008 [27]; Goloboff & Morales 2023 [28]) employing the *chkmoves* algorithm. The following taxa were identified as rogue taxa in the original dataset: *Aptychotrema, Bathyraja,* †*Cobelodus,* †*Cyclobatis,* †*Eorhinobatos, Glaucostegus, Hypanus,* †*Iansan,* †*Lessiniabatis,* †*Ozarcus,* †*Ostarriraja, Pseudobatos,* †*Pseudorhinobatos, Raja, Rhinobatos,* †'*Rhinobatos*' *hakelensis,* †'*R.*' *whitfieldi,* †*Rhombopterygia,* †*Stahlraja,* †*Tlalocbatus,* and

*Urobatis*. For subsequent analyses, all fossil ingroup rogue taxa were removed, while extant and outgroup rogue taxa were retained. The rationale behind this was to establish a balance between comprehensive taxon and outgroup sampling and reducing the risk of compromising phylogenetic accuracy by excluding taxa that were not relevant to our analyses. In addition, the Jurassic shark †*Protospinax annectans* was excluded from subsequent analyses due to its ambiguous phylogenetic position (see Jambura et al., 2023 [29]). The character states for the Jurassic batomorphs †*Asterodermus platypterus*, †*Belemnobatis sismondae*, and †*Spathobatis bugesiacus* were revised based on the latest morphological description of these taxa by Türtscher et al. (2024) [4]. In addition, the Jurassic rays †*Aellopobatis bavarica* and the newly described species †*Apolithabatis seioma* gen. et sp. nov. were included in the data matrix. The final version of the modified character matrix contained 76 taxa (44 extant and 32 fossil taxa) and 142 morphological characters, and was compiled in Mesquite 3.81 (Maddison & Maddison, 2023 [30]; S5 File

A parsimony analysis was conducted on the modified dataset using the command-line version of TNT 1.6 on MacOS (Goloboff et al., 2008 [27]; Goloboff & Morales, 2023 [28]). A traditional (heuristic) search was performed with 10.000 replicates of random stepwise addition, tree bisection and reconnection (TBR) branch swapping, and 100 trees stored per replicate. All most parsimonious trees (MPT) were used to compute consensus trees. Bootstrap and jackknife frequencies were calculated from 1.000 replicates under a traditional search with default settings.

A maximum likelihood (ML) analysis was conducted using the free online phylogenetic tool W-IQ-TREE (Trifinopoulos et al., 2016 [31]). W-IQ-TREE utilizes a dedicated computer cluster at the University of Vienna and is based on the latest version of IQ-TREE (Nguyen et al., 2015 [32]). Morphology was set as the sequence type, and the substitution model was automatically determined by the built-in ModelFinder (Kalyaanamoorthy et al., 2017 [33]). Branch support was calculated from 1.000 replicates using ultrafast bootstrapping (Hoang et al., 2018 [34]).

Log files for all analyses are available in S8 File.

## Systematic palaeontology

Class CHONDRICHTHYES Huxley, 1880 [35]
 Subclass ELASMOBRANCHII Bonaparte, 1838 [36]
 Cohort EUSELACHII Hay, 1902 [37]
 Subcohort NEOSELACHII Compagno, 1977 [38]
 Superorder BATOMORPHII Cappetta, 1980 [39]
 †APOLITHABATIFORMES ord. nov.

**Nomenclatural remarks.** Late Jurassic holomorphic batomorphs show a surprising resemblance to modern Rhinopristiformes, especially to present-day guitarfishes (Rhinobatidae, Glaucostegidae) and wedgefishes (Rhinidae). Due to this superficial resemblance, they traditionally were assigned to the family Rhinobatidae (e.g., Zittel, 1887–1890 [40]; Woodward, 1889a [41]; 1889b [42]; Thies, 1995 [43]; Underwood & Ward, 2004 [44]; Kriwet et al., 2009 [45]; Thies & Leidner, 2011 [46]; Klug & Kriwet, 2013 [47]). After Dames (1888) [48] first introduced the family †Spathobatidae, it was essentially forgotten until Underwood (2006) [6] used the family to include all genera of Late Jurassic batomorphs known from holomorphic fossils, but without specifying to which order this family belongs, since there was little evidence at that time of its position relative to other clades. Recent phylogenetic studies based on morphological characters (Villalobos-Segura et al., 2019 [49]; 2022 [26]; this study [see 'Phylogenetic analysis' below]) and on a combination of molecular and morphological data (Jambura et al., 2023 [29]) posited that Late Jurassic batomorphs form a monophyletic group when

maximum parsimony was used as optimality criterion. We follow this hypothesis here, although it should be noted that analyses of the same data sets with different optimality criteria (i.e., maximum likelihood and Bayesian inference) resulted in conflicting topologies and do not support a monophyletic group of Late Jurassic batomorphs (see Villalobos-Segura & Underwood, 2020 [50]: Fig 1; Villalobos-Segura et al., 2022 [26]: Figs 50 and 51; Jambura et al., 2023 [29]: S2 Fig in S3 File). According to the hypothesis accepted here, these batomorphs are sister to all remaining batomorphs and thus cannot be assigned to the family Rhinobatidae.

The monophyletic arrangement of these Late Jurassic batomorphs is only defined by few synapomorphies, but these characters unambiguously support their monophyly. The most important character uniting this group is the mesopterygium, which is elongated, follows the outline of the propterygium anteriorly to some extent, and thus is relatively large and similar in shape to the propterygium (see Villalobos-Segura et al., 2019 [49] [character 93], Villalobos-Segura et al., 2022 [26] [character 106], Jambura et al., 2023 [29] [character 110], and this study [character 81]). A character that is only found in Rhinopristiformes is the posterolateral connection of the antorbital cartilage with the nasal capsule, which is positioned, conversely, lateral in Late Jurassic batomorphs (see Villalobos-Segura et al., 2022 [26] [character 110], de Carvalho, 2004 [51] [character 2], Jambura et al., 2023 [29] [character 24], and this study [character 20]). Additionally, Late Jurassic taxa are the only members of Batomorphii (other than the undescribed batomorph from the Toarcian of Holzmaden, Germany) in which true dorsal fin spines are present (although absent in some species), thus showing retention of a plesiomorphic chondrichthyan character absent in crown-group batomorphs (see Villalobos-Segura et al., 2022 [26] [character 130], Jambura et al., 2023 [29] [character 177], and this study [character 134]).

We consequently propose a new order, †Apolithabatiformes ord. nov., which includes the single family, †Spathobatidae with the genera †*Aellopobatis*, †*Apolithabatis* gen. nov., †*Asterodermus*, †*Belemnobatis*, †*Kimmerobatis*, and †*Spathobatis*. This order represents the most plesiomorphic clade within Batomorphii being placed on the stem of the total group Batomorphii. It is possible that the hitherto undescribed Late Jurassic batomorph from Argentina, previously considered merely as Batomorphii indet. by Cione (1999) [14], also belongs to this order, which, however, can only be clarified by a detailed examination of the specimen.

**Type species.** †*Apolithabatis seioma* gen. et sp. nov.

**Included taxa.** †*Aellopobatis bavarica*, †*Apolithabatis seioma* gen. et sp. nov., †*Asterodermus platypterus*, †*Belemnobatis* spp., †*Kimmerobatis etchesi*, †*Spathobatis* spp.

**Etymology.** The name 'Apolithabatiformes' is composed of two Greek words, i.e., 'απολθωμα' (apolíthoma) meaning 'fossil' and 'βατς' (batís) meaning 'ray' or 'skate'.

**Diagnostic characters.** Elongated mesopterygium contiguous with propterygium and similar in shape; lateral articulation of antorbital cartilages (if present) to nasal capsules; two true fin spines anterior to dorsal fins (absent in some taxa);

Family †SPATHOBATIDAE Dames, 1888 [48]

†*APOLITHABATIS* gen. nov.

**Type species.** †*Apolithabatis seioma* gen. et sp. nov.

**Included taxa.** Type species only.

**Etymology.** Identical to that of the order †Apolithabatiformes (see above): the genus name '*Apolithabatis*' is composed of two Greek words, i.e., 'Απολθωμα' (apolíthoma) meaning 'fossil', and 'βατς' (batís) meaning 'ray' or 'skate'.

**Stratigraphic and geographic distribution.** Only known from the upper Kimmeridgian (Upper Jurassic) of the 'Solnhofen Archipelago' (Painten), Bavaria, Germany.

**Diagnosis.** A guitarfish-like batomorph unique in having the following combination of characters: heart-shaped disc that is wider than long; pointed snout; antorbital cartilages present but reaching less than halfway between the nasal capsules and the propterygium; vertebral

centra extending less than half of the synarcual length; large mesopterygium tangent to the propterygium; 40 pectoral radials (9 propterygial, 11 mesopterygial, 20 metapterygial); no pectoral radials articulate directly with the scapulocoracoid between the meso- and metapterygium; pectoral radials segmented in up to five segments; at least 16 pairs of ribs; 19 basipterygial radials (including one compound radial); puboischiadic bar curved anteriorly; no postpelvic processes present; broad and triangular lateral prepelvic processes; well-developed and plate-like haemal and supraneural spines; conspicuous bulge-like structure formed by the supraneural spines in front of each dorsal fin; no fin spines present.

†*APOLITHABATIS SEIOMA* gen. et sp. nov.

(Figs 4–7, S1 Fig in S3 File)

urn:lsid:zoobank.org:act:B45AD171-B766-4D37-8E88-21FB2CF65F08

**Etymology.**  The species name '*seioma*' is derived from the Greek 'σεω' (seíǫ), a verb stemming from the noun 'σεισμός' (seismós), meaning 'shake' or 'jiggle', in reference to the blasting of the fossil from the rock.

**Holotype.**  DMA-JP-2010/007

**Referred material.**  Only known from the holotype (DMA-JP-2010/007).

**Locality.**  Quarry of Rygol Company near the village of Painten, Lower Bavaria, South Germany.

**Diagnosis.**  As for the genus.

## Description

**Body shape.**  Large-sized batomorph, reaching a total length of at least 120 cm. The body shape is guitarfish-like. The disc is large and heart-shaped, reaching a width of 48.4% of the total length. With a length of 47.2% TL, the disc is slightly wider than long. The tail is narrow and long and accounts for 61.3% TL, measured from the pelvic girdle to the tip of the caudal fin. The two dorsal fins are located well posterior to the pectoral girdle; the first dorsal fin originates at 63.3% TL, the second dorsal fin at 73.9% TL. Both dorsal fins are nearly equal in size, the first dorsal fin has a height of 4.8% TL and a width of 5.3% TL, the second dorsal fin has a height of 5.2% TL and a width of 5.6% TL (Fig 4 and S1 Fig in S3 File).

**Neurocranium.**  The neurocranium of DMA-JP-2010/007 is partly crushed and details posterior to the nasal capsules are difficult to discern. The rostrum is quite robust and tapers slightly in mid-length. The anterior-most portion of the rostral cartilage is covered with skin, but apparently extends to the tip of the snout. As such, the rostrum is longer than the remaining part of the neurocranium. The nasal capsules are almost oval and inclined anteriorly. The width of the nasal capsules and the jaw cartilages is almost identical. The antorbital cartilages are elongated with an indentation in the mid-region of the anterior edge. The connection to the postero-lateral parts of the nasal capsules is rather narrow. The antorbital cartilages reach less than half the distance between the nasal capsules and the propterygia (Fig 5A and 5B).

**Jaws and branchial skeleton.**  Large parts of the central part of the jaws are fractured, thus their complete shape is not recognizable. However, the distal parts, particularly the right distal part, are preserved. A small dorsal flange of the Meckel's cartilage hooking around the posterior part of the palatoquadrate is discernible. Both jaw cartilages are almost straight and equal in antero-posterior depth. Due to taphonomic damage, structures such as the basihyals and basibranchials are not clearly discernible. Anteriorly, the branchial arches are approximately as wide as the jaws, but the width of the branchial apparatus tapers toward the pectoral girdle. The ceratobranchials are long and narrow (Fig 5A–5D).

**Pectoral girdle and fins.**  The scapulocoracoid is broken in the middle, with the right side still in its natural position and the left side displaced upwards. All three basal cartilages

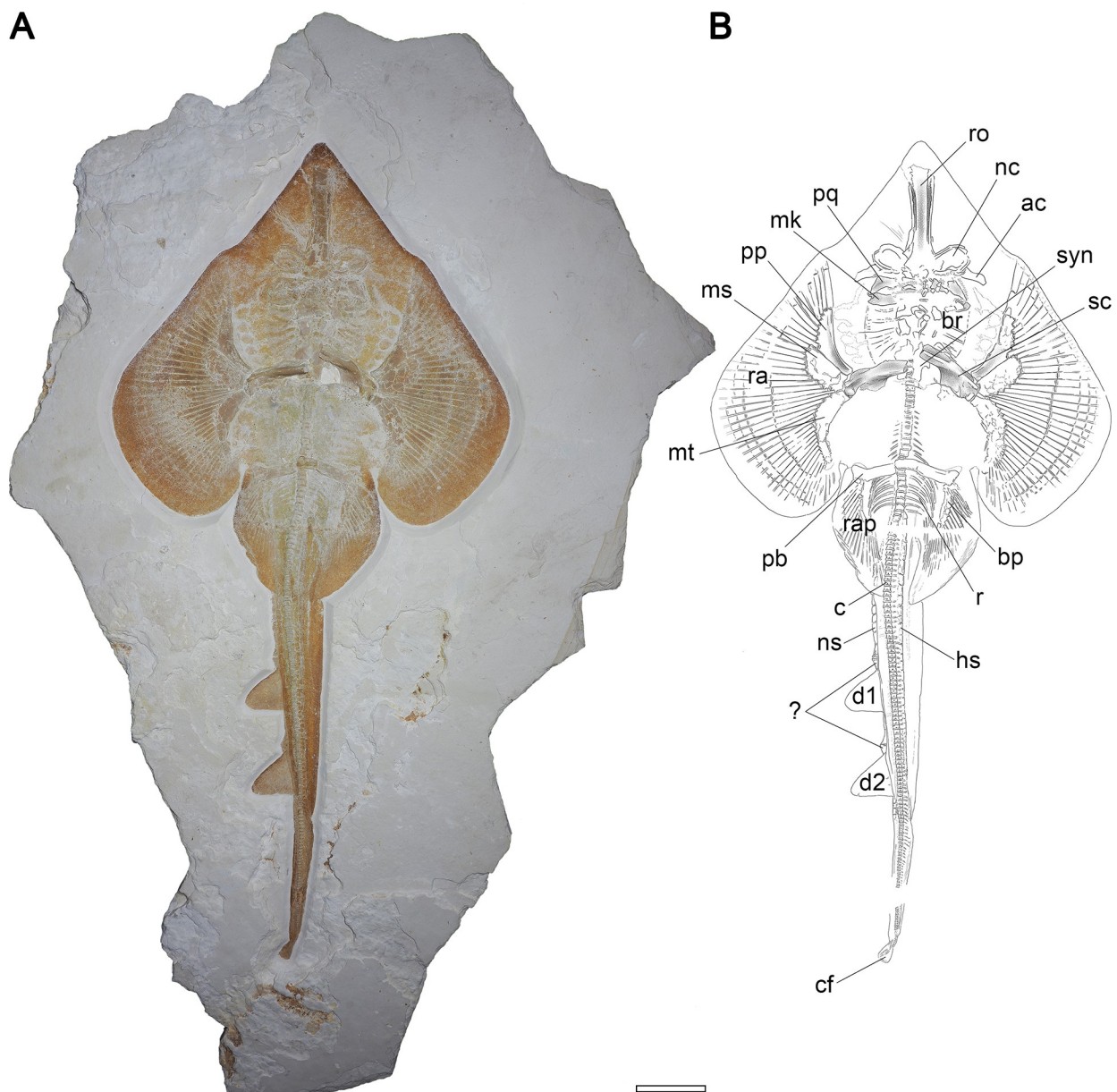

**Fig 4. Overview of DMA-JP-2010/007, the holotype of †*Apolithabatis seioma* gen. et sp. nov.** A) photograph of the specimen. B) Illustration of the specimen showing the skeletal morphology. Abbreviations: ac, antorbital cartilage; bp, basipterygium; br, branchial arches; c, vertebral centra; cf, caudal fin; d1, first dorsal fin; d2, second dorsal fin; hs, haemal spine; mk, Meckel's cartilage, ms, mesopterygium; mt, metapterygium; nc, nasal capsule; ns, neural spine; pb, puboischiadic bar; pp, propterygium; pq, palatoquadrate; r, ribs; ra, pectoral fin radials; rap, pelvic fin radials; ro, rostrum; sc, scapulocoracoid; syn, synarcual. The scale bar equals 10 cm.

articulate directly with the scapulocoracoid. The propterygium is broad and unsegmented. The mesopterygium is large, oval shaped, and adjacent to the propterygium for almost its entire length. The metapterygium is strongly curved and very broad in the first half, tapering increasingly in the second half. Of the three basal cartilages, the propterygium has the smallest articulation with the scapulocoracoid and the mesopterygium has the widest. In total, 40 radials articulate with the basal cartilages (9 with the propterygium, 11 with the mesopterygium, and 20 with the metapterygium). No radial articulates directly with the scapulocoracoid. At the widest part of the disc, the radials are composed of five elements (Fig 6A and 6B).

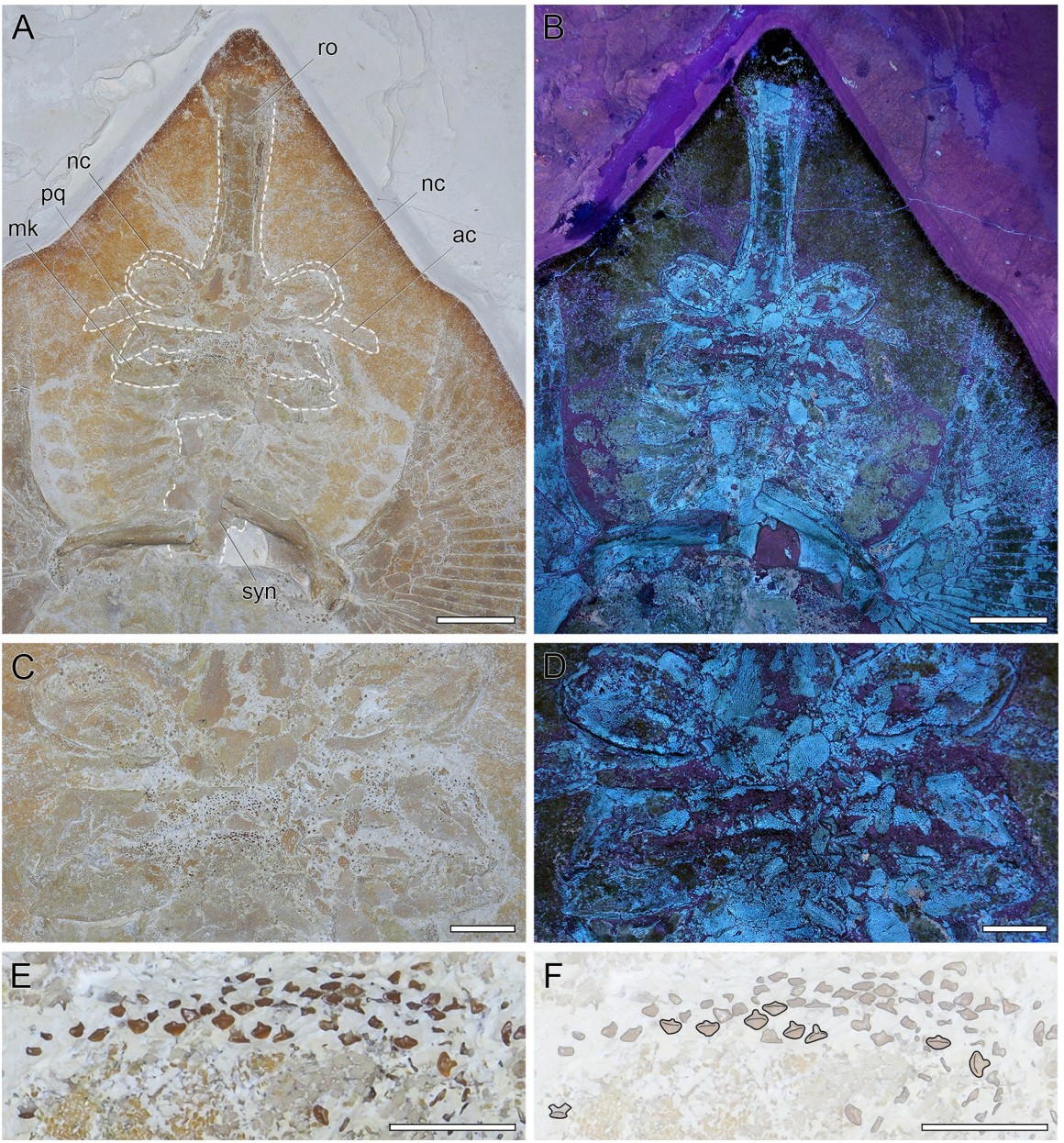

**Fig 5. Head region of DMA-JP-2010/007.** A) Close-up of the head region with several skeletal structures outlined. Abbreviations: ac, antorbital cartilage; mk, Meckel's cartilage; nc, nasal capsule; pq, palatoquadrate; ro, rostrum. B) Close-up of the head region under ultraviolet light. C) Close-up of the nasal capsules and the jaw cartilages under normal light and D) under ultraviolet light. E-F) Close-up of the teeth. Scale bars: A-B) 5 cm, C-D) 2 cm, E-F) 0.5 cm.

**Axial skeleton and unpaired fins.** The vertebral column consists of cyclospondylic centra that gradually decrease in size along the body. The exact outline of the synarcual is obscure due to taphonomic damage to the head region, but it appears to be broad at the insertion to the neurocranium, tapers in the middle, then widens and forms distinct lateral stays, and then tapers again. The vertebral centra extend less than half of the length of the synarcual. Sixteen pairs of ribs are present. No fin spines are associated with the dorsal fins. The well-developed and plate-like haemal spines seem to appear just after the last pair of ribs and are present throughout the tail. While the specimen is preserved in ventral view, the tail is rotated around

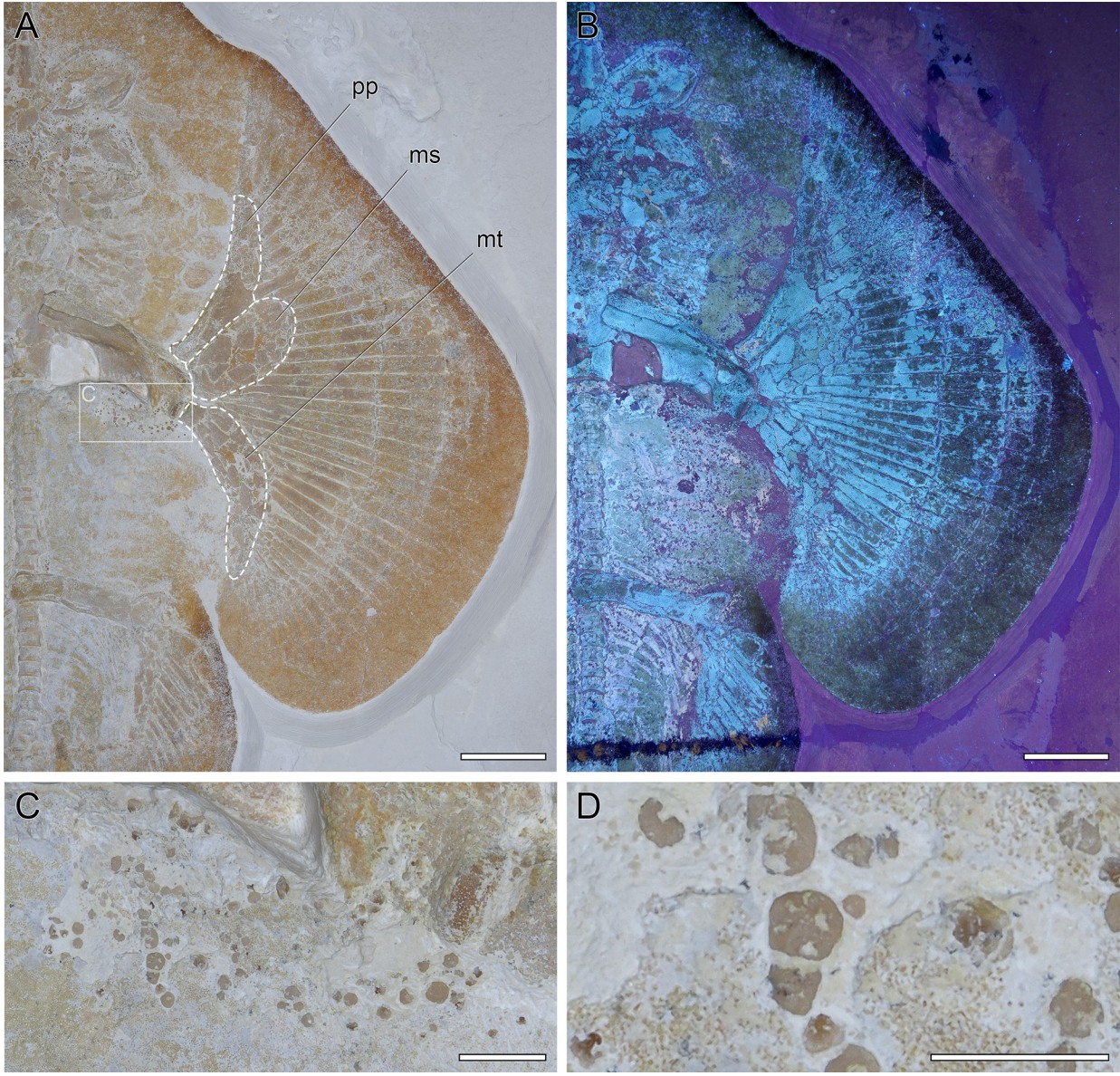

**Fig 6. Left pectoral fin of DMA-JP-2010/007.** A) Close-up of the left pecotral fin with the basal cartilages outlined. Abbreviations: ms, mesopterygium; mt, metapterygium; pp, propterygium. B) Close-up under ultraviolet light. C) Close-up of a patch of irregularly shaped denticles. D) Magnification of some of the denticles. Scale bars: A-B) 5 cm, C) 1 cm, D) 0.5 cm.

its longitudinal axis, best seen caudally of the second dorsal fin by a prominent fold. The bases of the dorsal fins (as well as that of the caudal fin) are thus somewhat obscured and not fully visible. Due to taphonomic processes, the dorsal fins appear to have been flipped over first, followed by the rigid spine. This apparently resulted in the supraneural spines piercing the already decomposing skin, and the flipped dorsal fins becoming 'stuck' between the supraneural spines and the folded skin on the right side of the tail. The supraneural spines, only visible in the tail, are well-developed and plate-like. Just anterior to each dorsal fin, the supraneural spines form a bulge-like structure. Since there are no traces of possible fin spines, we interpret this structure as a fixation point of connective tissue, acting as a reinforcement of the leading edge. In the living animal, these structures may well have been positioned directly

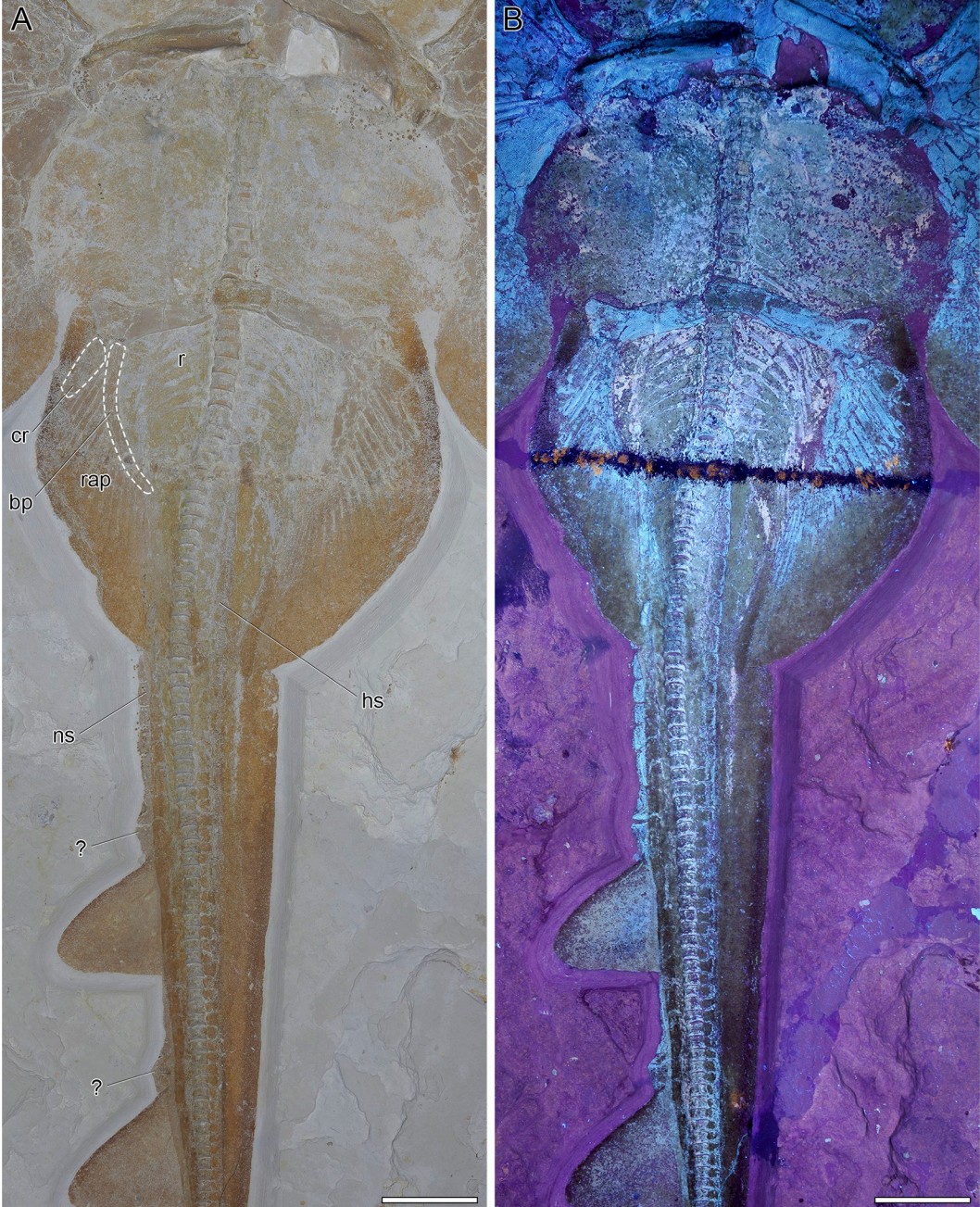

**Fig 7. Pelvic region of DMA-JP-2010/007.** A) Close-up with several skeletal structures outlined. Abbreviations: bp, basipterygium; cr, compound radial; hs, haemal spine; ns, neural spine; r, ribs; rap, pelvic fin radials. B) Close-up under ultraviolet light. Scale bars equal 5 cm.

at the base of each dorsal fin, but since the skin appears to have detached and shifted after the animal's death, it is possible that the dorsal fins are slightly displaced caudally (Fig 7A and 7B).

**Pelvic girdle, fins, and claspers.** Specimen DMA-JP-2010/007 is female, therefore no claspers can be observed. The pelvic fins reach 15.6% TL and are more than twice as long as wide. The basipterygium is slightly curved and narrow. One large and robust compound radial as well as 18 radials articulate with the basipterygium. The puboischiadic bar is not straight but

curved anteriorly; in the middle it shows an additional incurvation. The lateral prepelvic processes are very broad and triangular (Fig 7A and 7B).

**Teeth and denticles.**   The teeth are wider mesiodistally than labio-lingually deep. A mesiodistal ridge separates the labial face from the lingual face of the crown. A labial apron is absent or highly reduced and barely detached from the crown. The lingual uvula is narrow and long. The root appears to be holaulacorhize, with a distinct groove dividing it into two well-developed lobes (Fig 5E and 5F).

Irregularly shaped dermal denticles are present in the region of the nasal capsules, along the pectoral girdle (Fig 6C and 6D), along a short part of the vertebral column anterior to the pelvic girdle, and anterior to the first dorsal fin.

**Comparison.**   The snout has no knob-like or paddle-shaped projection, contrary to the condition seen in †*Ae. bavarica*, †*As. platypterus*, and †*S. bugesiacus*, but similar to what is seen in †*K. etchesi*. The antorbital cartilages reach less than halfway the distance between the nasal capsules and the propterygium, similar to the condition in †*Ae. bavarica*, †*S. bugesiacus*, and †*K. etchesi* (in †*As. platypterus*, the antorbital cartilages extend [usually less than halfway, but] up to halfway between the nasal capsules and the propterygium). The antorbital cartilages have straight posterior and anterior margins with an indentation in the middle, contrary to what is developed in †*Ae. bavarica* [both margins straight], †*As. platypterus* [no indentations but a posteriorly curved shape], and †*K. etchesi* [triangular in shape].; In the new taxon, no pectoral radials articulate directly with the scapulocoracoid between the meso- and metapterygium, which is different in †*S. bugesiacus*, but resembles the condition in †*Ae. bavarica*, †*As. platypterus*, and †*K. etchesi*. Conversely to †*As. platypterus* and †*S. bugesiacus*, †*Ap. seioma* gen. et sp. nov. has no fin spines preceeding the dorsal fins, which is also the case in †*Ae. bavarica*.**Results**

## Traditional morphometrics

**General.**   Using the new data set (Fig 2, S1 Table in S2 File), we performed a Principal Component Analysis considering all measurements that resulted in 25 axes (S2 Table in S2 File), with the first four each explaining more than 5% of the variation and together accounting for 77.89% of the total variability (see S3 Table in S2 File for loading values). The most important variables for PC1 and PC2 are described in detail later in this section (see below). The occupied morphospace plotted on the first two axes and the associated variables are shown in Fig 8.

Along PC1 (38.3% of the variation), †*Ae. bavarica*, †*As. platypterus*, and †*S. bugesiacus* are well separated, while †*Ap. seioma* gen. et sp. nov. overlaps with †*Ae. bavarica*. The occupied morphospaces of †*As. platypterus* and †*S. bugesiacus* are both in the positive range of PC1, while those of †*Ae. bavarica* and †*Ap. seioma* gen. et sp. nov. are in the negative range. This is also reflected in the most important variables of PC1, which mostly overlap in †*Ae. bavarica* and †*Ap. seioma* gen. et sp. nov., as well as in †*As. platypterus* and †*S. bugesiacus* (see below; see also Figs 8B and 9). Along PC2 (24.44% of the variation), †*Ae. bavarica* and †*Ap. seioma* gen. et sp. nov. can be clearly distinguished. †*Asterodermus platypterus* overlaps slightly with †*Ae. bavarica*, and †*S. bugesiacus* overlaps with both †*Ae. bavarica* and †*As. platypterus*. †*Apolithabatis seioma* gen. et sp. nov. does not overlap with any other occupied morphospace along PC2. Both †*Ae. bavarica* and †*S. bugesiacus* are predominantly in the positive range of PC2, †*As. platypterus* is almost entirely in the negative range, and †*Ap. seioma* gen. et sp. nov. is located in the negative morphospace of PC2. Within the main variables of PC2, †*Ap. seioma* gen. et sp. nov. overlaps only slightly with the other taxa, but separates most clearly from †*Ae. bavarica* (see below; see also Figs 8B and 9). The morphospaces plotted on PC1 and PC3

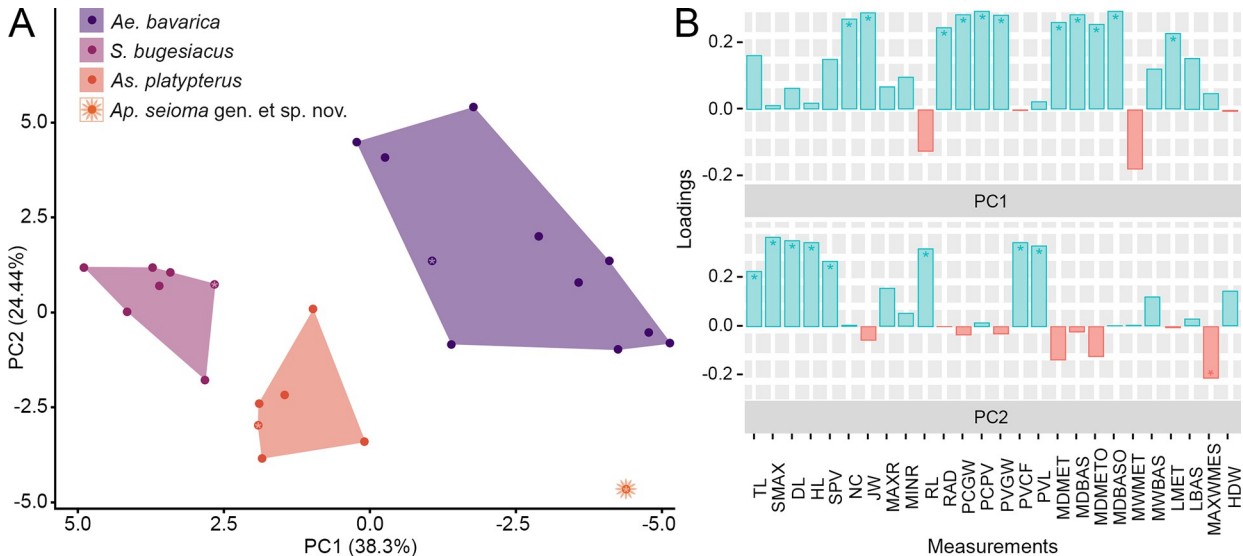

**Fig 8. Traditional morphometrics: Results of the principal component analysis (PCA), with each measurement adjusted to percentage of the disc width (DW) of each individual.** A) morphospace plotted on PC1 (38.3% of the total variance) and PC2 (24.44%). Asterisks indicate the holotype of the respective species. B) loading values showing the variables associated with the first two PC axes. Asterisks indicate the variables that explain the most variation for the respective PC axes. Abbreviations: DL, disc length; HDW, half disc width; HL, head length; JW, jaw width; LBAS, length of basipterygia; LMET, length of metapterygia; MAXR, maximum rostrum width; MAXWMES, maximum width of mesopterygia; MDBAS, inner maximum distance between basipterygia; MDBASO, outer maximum distance between basipterygia; MDMET, inner maximum distance between metapterygia; MDMETO, outer maximum distance between metapterygia; MINR, minimum rostrum width; MWBAS, maximum width of basipterygia; MWMET, maximum width of metapterygia; NC, nasal capsules maximum width; PCGW, pectoral girdle width; PCPV, pectoral girdle to pelvic girdle; PVCF, pelvic girdle to caudal fin tip; PVGW, pelvic girdle width; PVL, pelvic fin length; RAD, span between anteriormost fin radials; RL, rostrum length; SMAX, distance from the tip of the snout to the point of maximum disc width; SPV, snout to pelvic girdle; TL, total length.

(8.61% of the total variation) as well as on PC1 and PC4 (6.54% of the variation) along with the associated variables are shown in S2 Fig in S3 File.

The results of the Shapiro-Wilk normality test showed that *ca.* 84.62% of all measurements are normally distributed (i.e., 22 out of 26 measurements; S4 Table in S2 File and S4, S5 Figs in S4 File). The non-normally distributed measurements were further analyzed with the Kruskal–Wallis rank sum test, which indicates significant differences among the taxa for all but one measurement (i.e., total length; S5 Table in S2 File), which is further supported by Wilcoxon pairwise comparisons between the taxa (S6 Table in S2 File).

ANOVA tests on each normally distributed measurement showed significant differences in all but three measurements (i.e., maximum rostrum width, minimum rostrum width, pelvic girdle to caudal fin tip, maximum width of basipterygia, length of basipterygia, maximum width of mesopterygium, half disc width; S7 Table in S2 File), which was also largely confirmed by the Tukey test for pairwise comparisons (S8 Table in S2 File).

**Strongest vectors for PC1.** The following measurements were identified as having the strongest impact along PC1 and are compared interspecifically below: nasal capsules maximum width (NC), jaw width (JW), pectoral girdle width (PCGW), pelvic girdle width (PVGW), pectoral girdle to pelvic girdle (PCPV), length of metapterygia (LMET), maximum outer distance between metapterygia (MDMETO), maximum inner distance between metapterygia (MDMET), maximum outer distance between basipterygia (MDBASO), maximum inner distance between basipterygia (MDBAS), and span between last fin radials (RAD).

The nasal capsule maximum width (NC) amounts to 24.2% DW in †*Ap. seioma* gen. et sp. nov. and ranges from 20.7 to 28.9% DW in †*Ae. bavarica*, from 27.8 to 30.6% DW in †*As.*

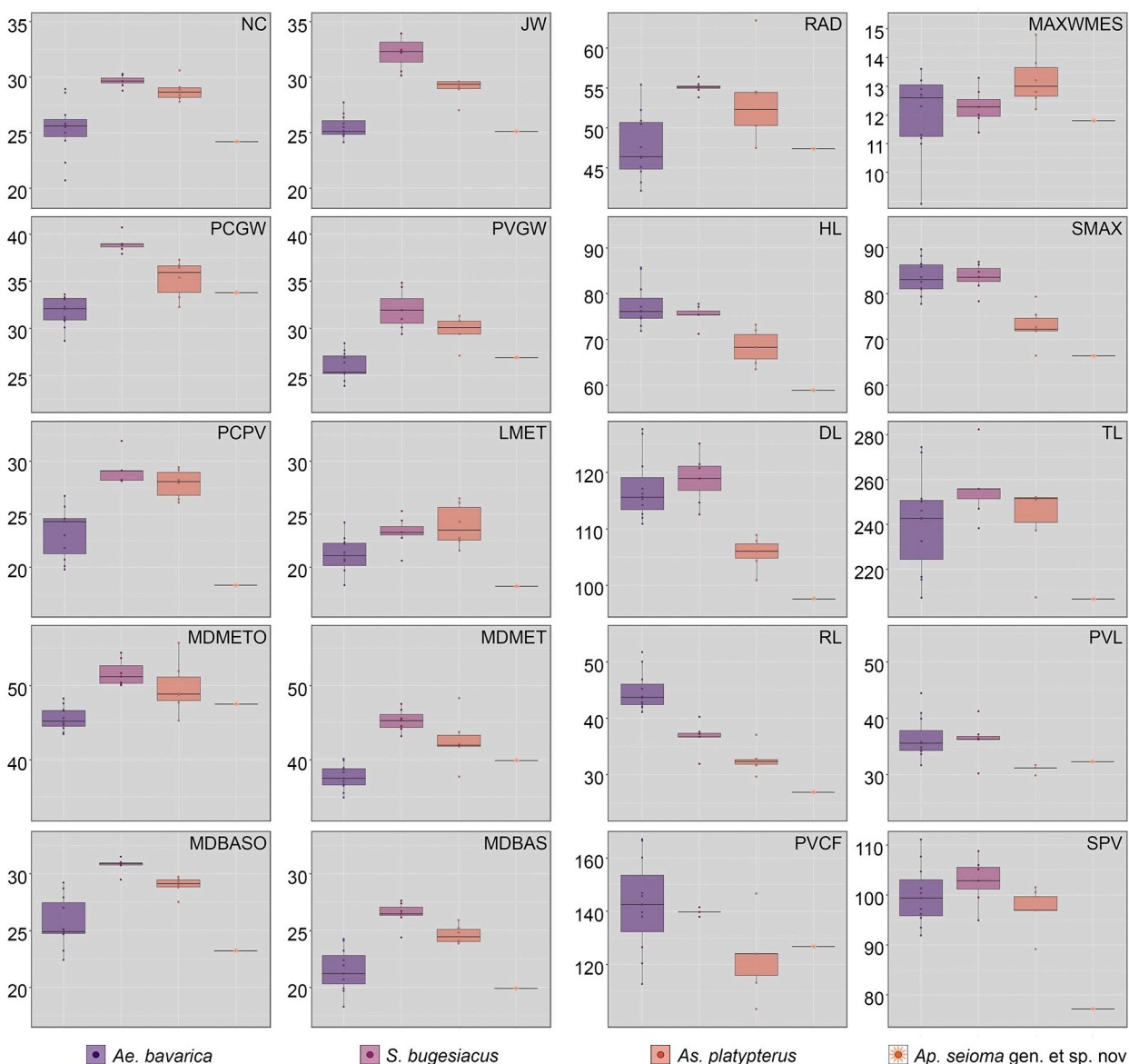

**Fig 9. Traditional morphometrics: Boxplots of the variables explaining the most variation of PC1 and PC2, respectively.** Measurements are adjusted as percentage of the disc width (DW). Abbreviations: DL, disc length; HL, head length; JW, jaw width; LMET, length of metapterygia; MAXWMES, maximum width of mesopterygia; MDBAS, inner maximum distance between basipterygia; MDBASO, outer maximum distance between basipterygia; MDMET, inner maximum distance between metapterygia; MDMETO, outer maximum distance between metapterygia; NC, nasal capsules maximum width; PCGW, pectoral girdle width; PCPV, pectoral girdle to pelvic girdle; PVCF, pelvic girdle to caudal fin tip; PVGW, pelvic girdle width; PVL, pelvic fin length; RAD, span between anteriormost fin radials; RL, rostrum length; SMAX, distance from the tip of the snout to the point of maximum disc width; SPV, snout to pelvic girdle; TL, total length.

*platypterus*, and from 28.8 to 32.9% DW in †*S. bugesiacus*. Statistically significant differences were found between †*Ae. bavarica* and †*S. bugesiacus* (S6 Table in S2 File, Fig 9).

Jaw width (JW) is 25.1% DW in †*Ap. seioma* gen. et sp. nov. and ranges from 24.1 to 27.7% DW in †*Ae. bavarica*, from 27.0 to 29.1% DW in †*As. platypterus*, and from 30.1 to 33.9% DW in †*S. bugesiacus*. The pairwise comparison shows that †*Ae. bavarica*, †*As. platypterus*, and †*S. bugesiacus* are all significantly different to each other in this relative measurement (S6 Table in S2 File, Fig 9).

The pectoral girdle width (PCGW) accounts for 33.8% DW in †*Ap. seioma* gen. et sp. nov. and ranges from 28.7 to 33.6% DW in †*Ae. bavarica*, from 32.3 to 37.3% DW in †*As. platypterus*, and from 37.9 to 40.7% DW in †*S. bugesiacus*. Statistically relevant differences were found between all taxa except between †*Ap. seioma* gen. et sp. nov. and †*Ae. bavarica* as well as †*As. platypterus* (S8 Table in S2 File, Fig 9).

The pelvic girdle width (PVGW) is 26.9% DW in †*Ap. seioma* gen. et sp. nov. and spans from 23.9 to 28.4% DW in †*Ae. bavarica*, from 27.1 to 31.3% DW in †*As. platypterus*, and from 29.4 to 34.8% DW in †*S. bugesiacus*. The largest differences were found between †*Ae. bavarica* and †*S. bugesiacus*, †*Ae. bavarica* and †*As. platypterus*, and between †*Ap. seioma* gen. et sp. nov. and †*S. bugesiacus* (S8 Table in S2 File, Fig 9).

The distance from the pectoral girdle to the pelvic girdle (PCPV) is 18.3% DW in †*Ap. seioma* gen. et sp. nov. and spans from 19.8 to 26.7% DW in †*Ae. bavarica*, from 26.1 to 29.4% DW in †*As. platypterus*, and from 28.1 to 31.9% DW in †*S. bugesiacus*. With the exception of †*Ae. bavarica* to †*Ap. seioma* gen. et sp. nov. and †*As. platypterus* to †*S. bugesiacus*, all taxa are statistically significantly different from each other (S8 Table in S2 File, Fig 9).

The length of the metapterygia (LMET) is 18.2% DW in †*Ap. seioma* gen. et sp. nov. and ranges from 18.3 to 24.2% DW in †*Ae. bavarica*, from 21.6 to 26.5% DW in †*As. platypterus*, and from 20.6 to 25.3% DW in †*S. bugesiacus*. No statistically significant differences were detected between the groups (S8 Table in S2 File, Fig 9).

The maximum outer distance between the metapterygia (MDMETO) amounts to 47.5% DW in †*Ap. seioma* gen. et sp. nov. and ranges from 43.6 to 48.2% DW in †*Ae. bavarica*, from 45.3 to 55.7% DW in †*As. platypterus*, and from 50.1 to 54.4% DW in †*S. bugesiacus*. Similarly, the maximum inner distance between the metapterygia (MDMET) amounts to 39.9% DW in †*Ap. seioma* gen. et sp. nov. and ranges from 34.9 to 40.1% DW in †*Ae. bavarica*, from 37.7 to 48.3% DW in †*As. platypterus*, and from 43.2 to 47.5% DW in †*S. bugesiacus*. Both pairwise comparisons reveal significant differences between †*Ae. bavarica* and †*As. platypterus* as well as between †*Ae. bavarica* and †*S. bugesiacus* (S8 Table in S2 File, Fig 9).

The maximal outer distance between basipterygia (MDBASO) accounts for 23.2% DW in †*Ap. seioma* gen. et sp. nov. and ranges from 22.4 to 29.2% DW in †*Ae. bavarica*, from 27.5 to 29.7% DW in †*As. platypterus*, and from 29.5 to 31.5% DW in †*S. bugesiacus*. With the exception of †*Ap. seioma* gen. et sp. nov., all groups are statistically significantly different from each other (S6 Table in S2 File, Fig 9).

The intraspecific range of the maximum inner distance between basipterygia (MDBAS) is similar to the maximal outer distance between basipterygia (MDBASO); it amounts to 19.9% DW in †*Ap. seioma* gen. et sp. nov. and ranges from 18.3 to 24.2% DW in †*Ae. bavarica*, from 23.9 to 25.9% DW in †*As. platypterus*, and from 24.4 to 27.6% DW in †*S. bugesiacus*. However, unlike the results for MDBASO, the pairwise comparison reveals significant differences between all taxa except between †*Ae. bavarica* and †*Ap. seioma* gen. et sp. nov., as well as between †*As. platypterus* and †*S. bugesiacus* (S8 Table in S2 File, Fig 9).

The span between last fin radials (RAD) is 47.4% DW in †*Ap. seioma* gen. et sp. nov. and ranges from 42.2 to 55.4% DW in †*Ae. bavarica*, from 47.5 to 63.4% DW in †*As. platypterus*, and from 53.8 to 56.7% DW in †*S. bugesiacus*. †*Aellopobatis bavarica* differs significantly from †*As. platypterus* as well as from †*S. bugesiacus* (S8 Table in S2 File, Fig 9).

**Strongest vectors for PC2.** The following measurements were identified as having the greatest impact along PC2 and are compared interspecifically below: maximum width of mesopterygium (MAXWMES), head length (HL), snout to maximum disc width (SMAX), disc length (DL), total length (TL), rostrum length (RL), pelvic-fin length (PVL), pelvic girdle to caudal fin tip (PVCF), and snout to pelvic girdle (SPV).

The maximum width of the mesopterygium (MAXWMES) measures 11.8% DW in †*Ap. seioma* gen. et sp. nov. and ranges from 8.9 to 13.7% DW in †*Ae. bavarica*, from 12.2 to 14.8% DW in †*As. platypterus*, and from 11.4 to 13.3% DW in †*S. bugesiacus*. The taxa are not statistically different (S8 Table in S2 File, Fig 9).

The head length (HL) is 58.9% DW in †*Ap. seioma* gen. et sp. nov. and ranges from 71.9 to 85.6% DW in †*Ae. bavarica*, from 63.5 to 73.2% DW in †*As. platypterus*, and from 71.2 to 77.7% DW in †*S. bugesiacus*. With the exception of †*Ae. bavarica* to †*S. bugesiacus* and †*As. platypterus* to †*Ap. seioma* gen. et sp. nov., all taxa are significantly different to each other (S8 Table in S2 File, Fig 9).

The distance from the tip of the snout to the point of maximum disc width (SMAX) accounts for 66.4% DW in †*Ap. seioma* gen. et sp. nov. and ranges from 77.8 to 96.6% DW in †*Ae. bavarica*, from 66.5 to 79.3% DW in †*As. platypterus*, and from 78.3 to 86.8% DW in †*S. bugesiacus*. With the exception of †*Ae. bavarica* to †*S. bugesiacus* and †*As. platypterus* to †*Ap. seioma* gen. et sp. nov., all taxa are significantly different to each other (S8 Table in S2 File, Fig 9).

The disc length (DL) is 97.6% DW in †*Ap. seioma* gen. et sp. nov. and spans from 110.9 to 127.6% DW in †*Ae. bavarica*, from 101.0 to 108.9% DW in †*As. platypterus*, and from 112.6 to 125.1% DW in †*S. bugesiacus*. With the exception of †*Ae. bavarica* to †*S. bugesiacus* and †*As. platypterus* to †*Ap. seioma* gen. et sp. nov., all taxa are significantly different to each other (S8 Table in S2 File, Fig 9).

The total length (TL) measures 206.7% DW in †*Ap. seioma* gen. et sp. nov. and ranges from 207.2 to 274.4% DW in †*Ae. bavarica*, from 207.3 to 251.1% DW in †*As. platypterus*, and from 238.3 to 282.4% DW in †*S. bugesiacus*. The taxa are not statistically different in this relative measurement (S8 Table in S2 File, Fig 9).

The length of the rostrum (RL) is 26.9% in †*Ap. seioma* gen. et sp. nov. and ranges from 41.2 to 51.7% DW in †*Ae. bavarica*, from 29.7 to 37.1% DW in †*As. platypterus*, and from 31.9 to 40.2% DW in †*S. bugesiacus*. All taxa differ significantly from each other, except †*As. platypterus* from †*Ap. seioma* gen. et sp. nov. and †*As. platypterus* from †*S. bugesiacus* (S8 Table in S2 File, Fig 9).

The length of the pelvic fin (PVL) is 32.3% DW in †*Ap. seioma* gen. et sp. nov. and spans from 31.7 to 44.4% DW in †*Ae. bavarica*, from 28.9 to 32.9% DW in †*As. platypterus*, and from 30.2 to 41.2% DW in †*S. bugesiacus*. Statistically significant differences were found between †*Ae. bavarica* and †*As. platypterus* and between †*As. platypterus* and †*S. bugesiacus* (S8 Table in S2 File, Fig 9).

The distance from the pelvic girdle to the caudal fin tip (PVCF) is 126.7% DW in †*Ap. seioma* gen. et sp. nov. and ranges from 112.7 to 166.6% DW in †*Ae. bavarica*, from 103.2 to 146.9% DW in †*As. platypterus*, and from 137.9 to 141.5% DW in †*S. bugesiacus*. The taxa are not statistically different in this relative measurement (S8 Table in S2 File, Fig 9).

The distance from the tip of the snout to the pelvic girdle (SPV) is 77.2% DW in †*Ap. seioma* gen. et sp. nov. and spans from 91.9 to 111.1% DW in †*Ae. bavarica*, from 89.2 to 101.5% DW in †*As. platypterus*, and from 95.0 to 108.7% DW in †*S. bugesiacus*. In this relative measurement, †*Ap. seioma* gen. et sp. nov. is statistically different from all other taxa (S8 Table in S2 File, Fig 9).

## Geometric morphometrics

**Head outline.** A PCA was performed on the data matrix (S1 Table in S2 File), resulting in 21 axes, with the first two accounting for 86.98% of the total shape variation (S9 Table in S2 File). The remaining 19 axes each account for <5% of the total variation.

The morphospace occupation of the taxa plotted on PC1 (77.61% of the total morphological variation) and PC2 (9.37% of the variation) shows no overlap between taxa. †*Apolithabatis*

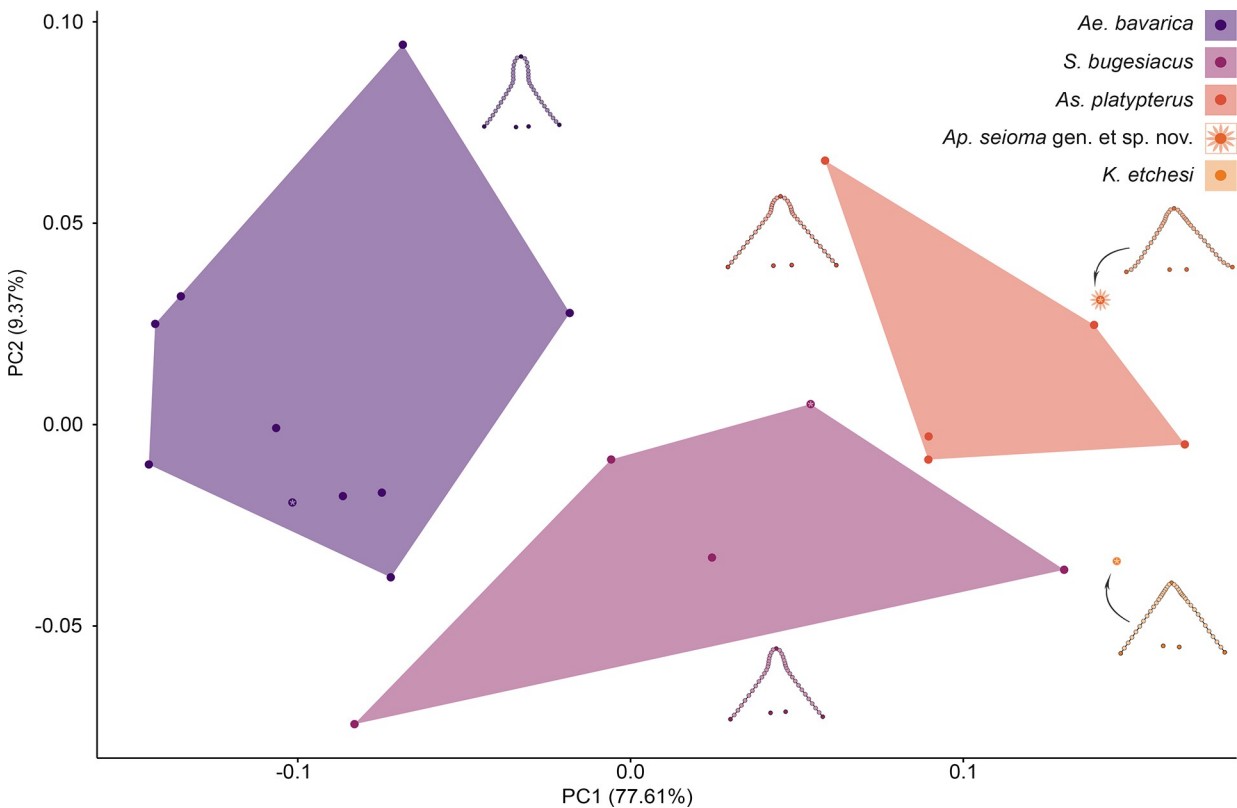

**Fig 10. Geometric morphometrics–head outline: Results of the principal component analysis (PCA).** Morphospace plotted on PC1 (77.61% of the total variation) and PC2 (9.37%). Asterisks indicate the holotype of the respective species. Mean shapes are shown next to each group. Dark-coloured dots of the mean shapes indicate true landmarks, light-coloured dots indicate semilandmarks.

*seioma* gen. et sp. nov. is positioned close to †*As. platypterus*, reflecting the similarly short rostra in comparison to †*Ae. bavarica* and †*S. bugesiacus*, which have elongated and paddle-shaped rostra. Likewise, †*K. etchesi* is located close between †*As. platypterus* and †*S. bugesiacus* on the morphospace (Fig 10).

PC1 describes the length and shape of the rostrum. †*Aellopobatis bavarica*, evidently the taxon with the most distinct rostrum, occupies the negative area of PC1, followed by †*S. bugesiacus*, which spans from negative to positive values of PC1. †*Asterodermus platypterus* is confined to the positive region of PC1. Both †*K. etchesi* and †*Ap. seioma* gen. et sp. nov. have a rather pointed head outline and lack a paddle-shaped rostrum which is reflected by their morphospace occupation in the positive area of PC1. The shape changes along PC2 are far more subtle and mainly comprise the width of the head (Fig 10).

A Procrustes ANOVA reveals significant differences in shape between the taxa ($R^2$ = 0.67184, F = 8.701, Z = 4.1144, *p* = 0.001; S10 Table in S2 File), which is further supported by a pairwise comparison (S11 Table in S2 File). The pairwise comparison shows that †*Ae. bavarica* is the most distinct taxon, with a significant difference to all others.

The second Procrustes ANOVA indicates significant differences in size between the taxa ($R^2$ = 0.75743, F = 13.271, Z = 3.0672, *p* = 0.001; S12 Table in S2 File). The pairwise comparison detects significant differences between †*K. etchesi* and †*Ae. bavarica*, †*As. platypterus*, as well as †*S. bugesiacus*. Furthermore, †*Ae. bavarica* and †*As. platypterus* also are significantly different in size (S13 Table in S2 File).

**Complete body.** Since only a few of the specimens are sufficiently preserved along the entire body to perform this landmark analysis, only 10 specimens were included here (S1 Table in S2 File). To capture the body outline but also the position of different skeletal structures, 11 landmarks and 136 semilandmarks were used on specimens of †*Ae. bavarica*, †*As. platypterus*, †*Ap. seioma* gen. et sp. nov., and †*S. bugesiacus*. Note the small sample size when interpreting the statistical results of this analysis.

The PCA resulted in 9 axes, with the first five accounting for 88.56% of the total variation. The remaining 4 axes each account for <5% of the total variation (S14 Table in S2 File).

When plotted on PC1 (36.96% of the total variation) and PC2 (23.66%), †*As. platypterus* and †*S. bugesiacus* are relatively close to each other. †*Aellopobatis bavarica* and †*Ap. seioma* gen. et sp. nov., on the other hand, are clearly separated from each other as well as from the other two taxa.

The negative area of PC1 is occupied by †*Ae. bavarica* and †*Ap. seioma* gen. et sp. nov., while †*As. platypterus* and †*S. bugesiacus* are restricted to the positive region of PC1. The negative values are associated with an elongation of the rostrum, resulting in the typical spatula-shaped rostrum seen in †*Ae. bavarica* in the most negative range. Also, the pelvic girdle slightly shifts forward and the pelvic gins become narrower, resulting in a quite slender caudal region. In addition, the base of the nasal capsules shifts slightly backwards and the head region becomes somewhat narrower (Fig 11).

In the positive region of PC1, the changes are reversed with the rostrum becoming shorter, the base of the nasal capsules moving forward, the head region becoming slightly wider, the

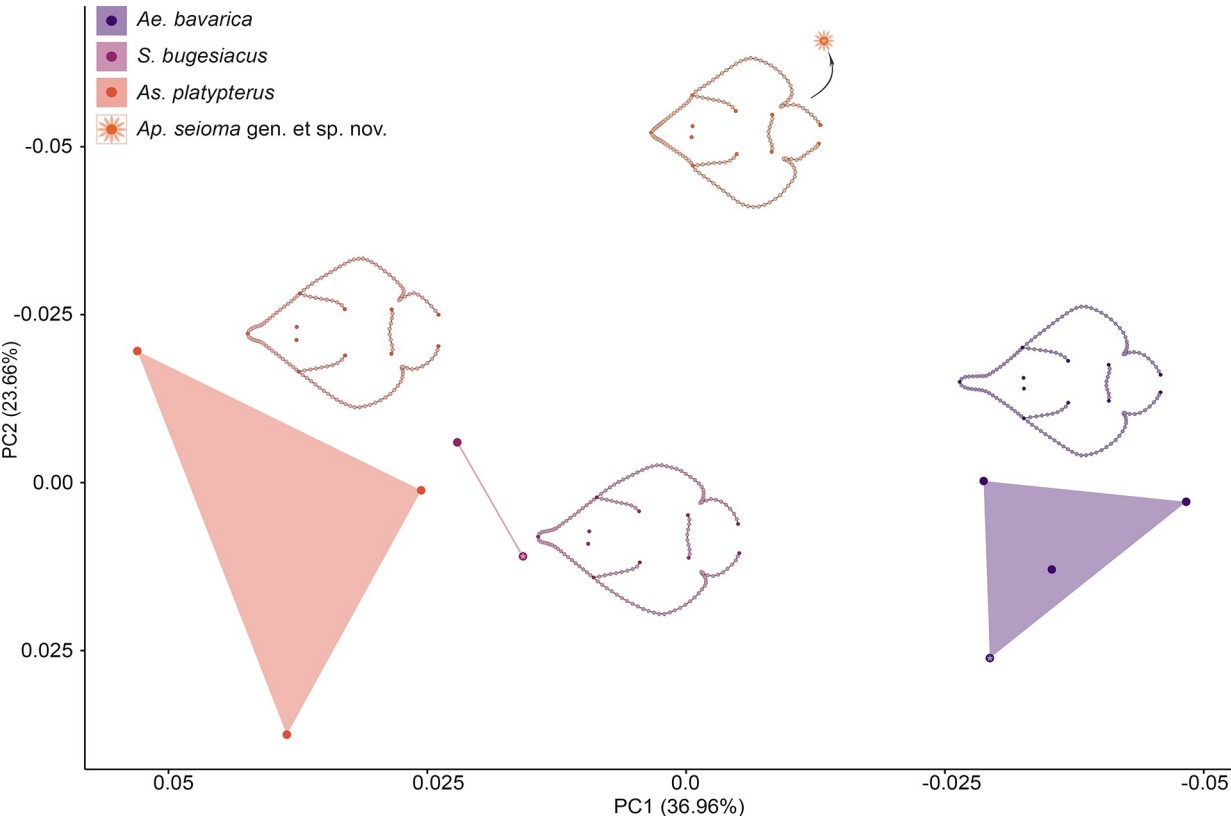

**Fig 11. Geometric morphometrics–complete body: Results of the principal component analysis (PCA).** Morphospace plotted on PC1 (36.96% of the total variation) and PC2 (23.66%). Asterisks indicate the holotype of the respective species. Mean shapes are shown next to each group. Dark-coloured dots of the mean shapes indicate true landmarks, light-coloured dots indicate semilandmarks.

pelvic girdle moving posteriorly, and the pelvic fins and caudal region increasing in width. The negative region of PC2 is characterized by a tapering of the rostrum. The head has a pointed shape and no spatula-shaped extension of the rostrum is present, neither short as seen in †*As. platypterus* and †*S. bugesiacus* nor prolonged as in †*Ae. bavarica*. In addition, the disc becomes much wider and the pelvic girdle shifts forward. In the positive region of PC2, the rostrum becomes slightly longer and the disc narrower. S3 Fig in S3 File depicts the morphospaces plotted on PC1 and PC2, PC3 (12.24%), PC4 (9.05%), and PC5 (6.64%), respectively. In all four morphospaces, the four taxa are clearly separated from each other and never overlap.

A Procrustes ANOVA reveals significant differences in shape between the taxa ($R^2$ = 0.61197, F = 3.1543, Z = 3.5893, $p$ = 0.001; S15 Table in S2 File). In the pairwise comparison, statistically significant differences are shown between †*Ae. bavarica* and †*Ap. seioma* gen. et sp. nov., †*Ae. bavarica* and †*As. platypterus*, as well as †*Ap. seioma* gen. et sp. nov.. and †*As. platypterus* (S16 Table in S2 File).

The detected differences in size between the taxa ($R^2$ = 0.74207, F = 5.7541, Z = 1.889, $p$ = 0.043; S17 Table in S2 File) is only supported by statistical significance for †*Ae. bavarica* and †*S. bugesiacus* in the pairwise comparison (S18 Table in S2 File).

### Phylogenetic analysis

The parsimony analysis resulted in 1611 most parsimonious trees (MPTs) with a best score of 371 steps. Consistency index (CI) and retention index (RI) of the MPTs were 0.461 and 0.857, respectively. Both maximum parsimony (MP) and maximum likelihood (ML) analyses indicated a basal phylogenetic placement of †*Ap. seioma* gen. et sp. nov. within the rays (superorder Batomorphii) and a sister relationship between †*Ap. seioma* gen. et sp. nov. and †*Ae. bavarica* (Fig 12 and S6 Fig in S9 File). Both analyses yielded conflicting results for the phylogenetic relationships among Jurassic rays, with the ML analysis providing a better-resolved phylogenetic tree (see Discussion below).

## Discussion

In the study by Türtscher et al. (2024) [4], five holomorphic Late Jurassic ray taxa were unambiguously confirmed, i.e. the early Tithonian taxa †*Aellopobatis bavarica*, †*Asterodermus platypterus*, and †*Kimmerobatis etchesi*, and the late Kimmeridgian taxa †*Belemnobatis sismondae* and †*Spathobatis bugesiacus*. †*Spathobatis*? *morinicus* from the lower Tithonian of Boulogne-sur-Mer (northern France) may represent another taxon, but needs to be studied and redescribed to confirm its affiliation with †*Spathobatis* and to determine whether it is indeed a separate species or belongs to †*Spathobatis bugesiacus*. It should be noted, however, that all skeletal remains of †*S. bugesiacus* so far have been found in the sedimentary deposits of Cerin (south-eastern France), which are of late Kimmeridgian age and therefore older than the early Tithonian Boulogne-sur-Mer deposit. Isolated teeth of †*Spathobatis bugesiacus*, however, have been reported from both Kimmeridigan and Tithonian aged outcrops in Europe (Underwood, 2002 [52]; Leuzinger et al., 2017 [53]). It is therefore likely that †*S. bugesiacus* was also present in the Tithonian, however, tooth-based species identifications should be treated with caution at this stage, as our knowledge of inter- and intraspecific variations in tooth morphology within Spathobatidae remains very incomplete (see Türtscher et al., 2024 [4]). The undescribed ray from the middle Tithonian of Argentina (Cione, 1999 [14]) may well represent another new taxon, which can only be confirmed after a detailed anatomical and taxonomical description. Here, we describe the sixth undoubtedly valid Late Jurassic ray taxon known from skeletal material, †*Apolithabatis seioma* gen. et sp. nov. from the 'Solnhofen Archipelago' in southern Germany. To the best of our knowledge, this taxon is the first batomorph from the late

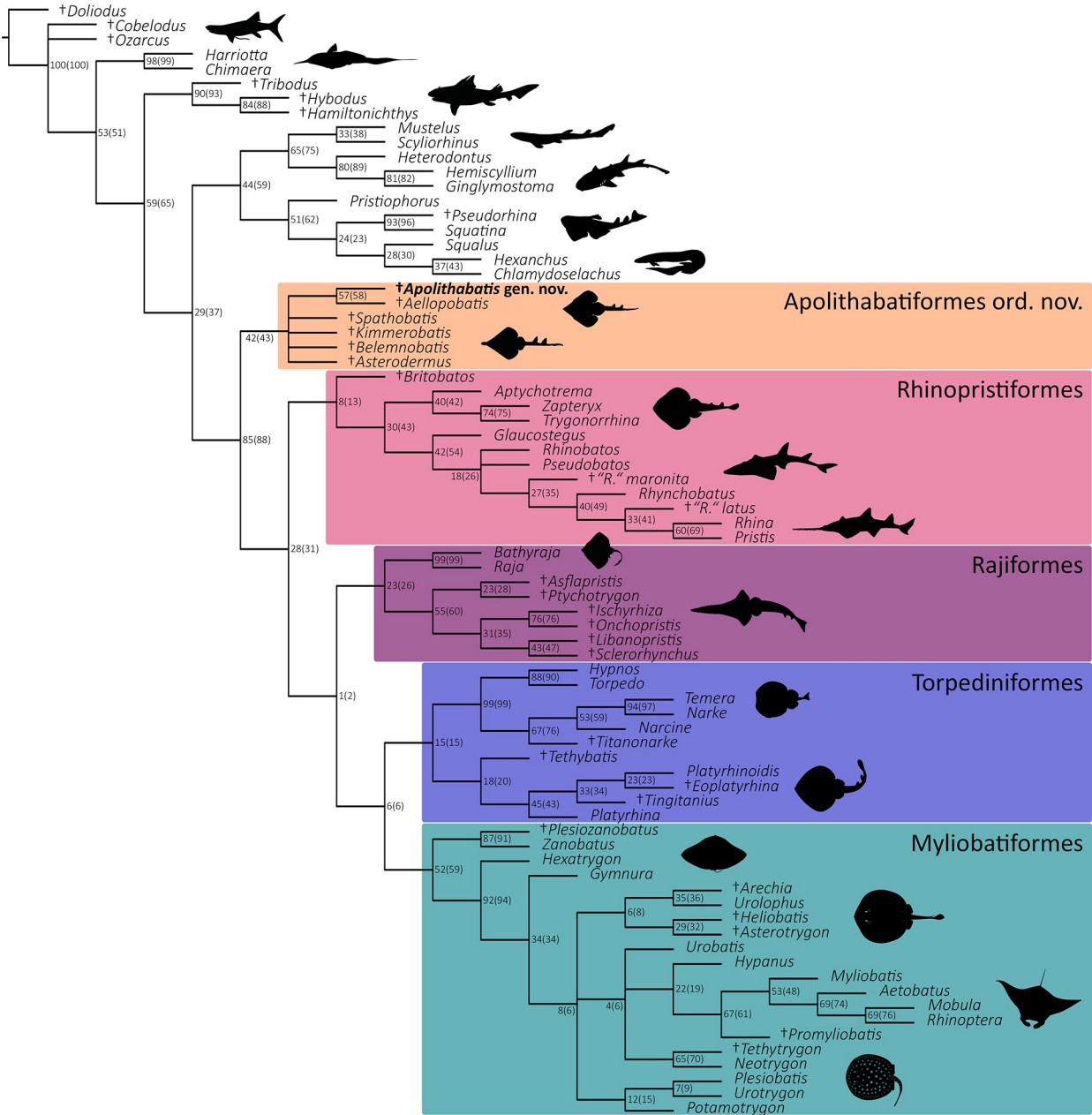

**Fig 12. Majority-rule consensus tree with bootstrap and jackknife frequencies (jackknife values in parentheses).** Daggers before taxon names indicate extinct taxa. Silhouettes were either drawn by the authors (JT, FS, PLJ) or downloaded from www.phylopic.org (all downloaded images were available for re-use under the Public Domain Dedication 1.0 licence).

Kimmeridgian of southern Germany and thus represents the oldest ray of the 'Solnhofen Archipelago' (for a palaeoenvironmental reconstruction of †*Apolithabatis seioma* gen. et sp. nov., see Fig 13).

The phylogenetic relationships within batomorphs have long been the subject of intense research, but despite a lot of progress in resolving various taxonomic and systematic issues, countless unanswered questions still persist. Therefore, including Jurassic ray taxa, which belong to the oldest known batomorph species to date, into phylogenetic analyses helps to generate better resolved hypotheses of batomorph evolution (Underwood & Claeson, 2019 [54]).

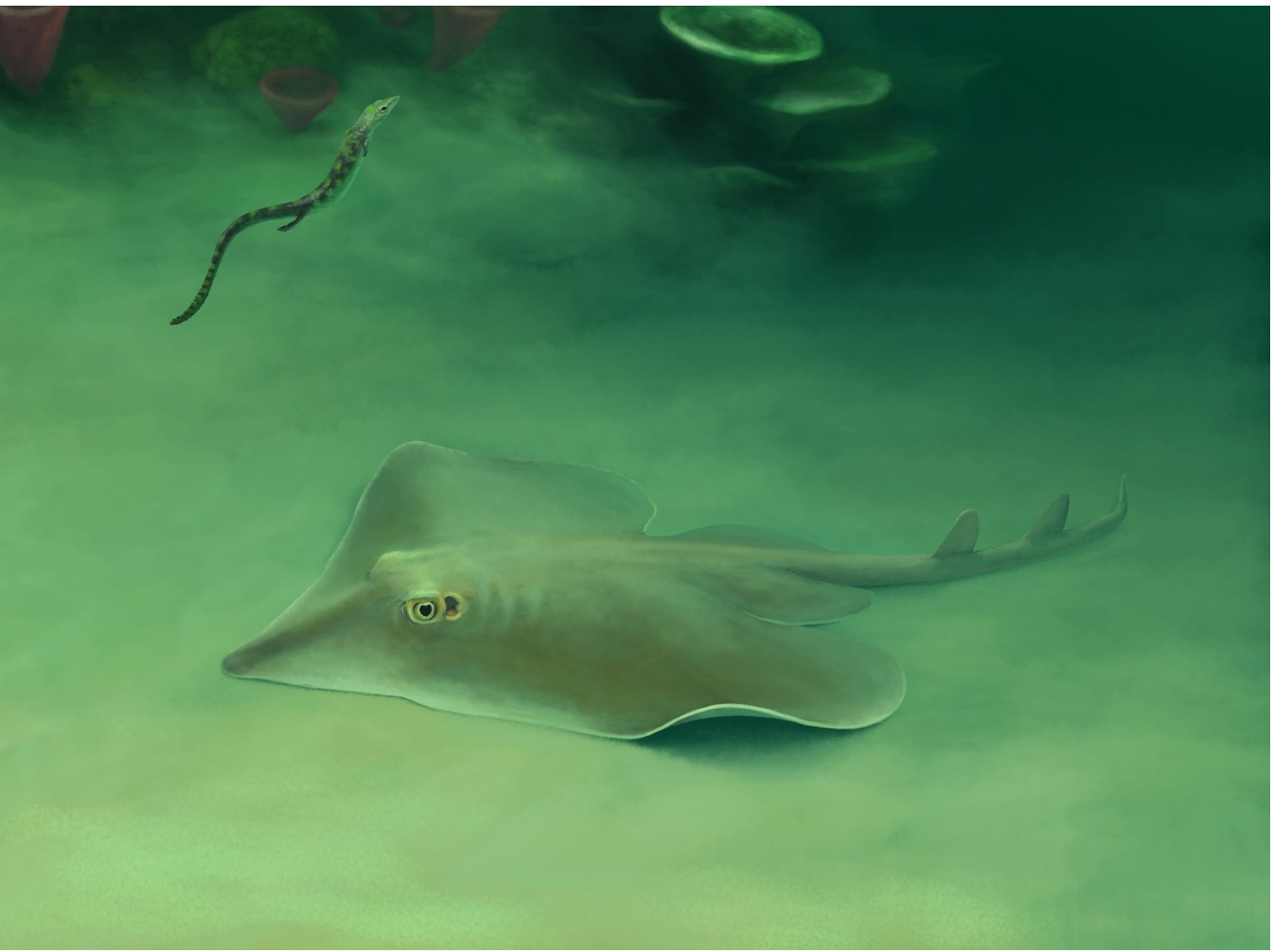

**Fig 13. Environmental reconstruction showing †***Apolithabatis seioma*** gen. et sp. nov. in association with a juvenile pleurosaurid.**

So far, several phylogenetic studies (Underwood & Claeson, 2019 [54]; Villalobos-Segura et al., 2019 [49]; 2022 [26]; Jambura et al., 2023 [29]) indicated that the Late Jurassic rays that are known from holomorphic specimens formed a monophyletic group, here named Apolithabatiformes ord. nov. In the phylogenetic study of Underwood and Claeson (2019) [54], these Late Jurassic batomorphs were placed within the crown-group batomorphs at the base of Rhinopristiformes, Myliobatiformes, Rajiformes, and platyrhinids. The order Torpediniformes was recovered as sister to all other batomorphs, including Apolithabatiformes ord. nov., within which †*K. etchesi* and †*B. sismondae* appear to be more closely related to each other than to †*S. bugesiacus*. However, in light of the recent results of Türtscher et al. (2024) [4], it should be noted that the alleged †*B. sismondae* specimen analyzed by Underwood and Claeson (2019) [54] actually belongs to †*As. platypterus* (CM 4408), and that the three supposed †*S. bugesiacus* specimens belong to †*As. platypterus* (NHMUK PV P 10934), †*Ae. bavarica* (CM 5396), and †*S. bugesiacus* (NHMUK PV P 2099), respectively. This, along with the meanwhile expanded data matrix, explains why the relationships within the Late Jurassic rays, as well as between these taxa and the remaining batomorphs as recovered by Underwood and Claeson (2019) [54], differ profoundly from those obtained by us. In the present study, five monophyletic

groups are recovered within the superorder Batomorphii, i.e., Myliobatiformes, Torpedini-formes, Rajiformes, Rhinopristiformes, and Apolithabatiformes ord. nov., which includes the genera †*Aellopobatis*, †*Apolithabatis* gen. nov., †*Asterodermus*, †*Belemnobatis*, †*Kimmerobatis*, and †*Spathobatis*. The new order Apolithabatiformes is recovered in a sister-group relation-ship with the remaining batomorphs (including Torpediniformes) and thus represents a mem-ber of stem group batomorphs. Similar results were obtained by Villalobos-Segura et al. (2022) [26], as well as by Villalobos-Segura et al. (2019) [49] and Jambura et al. (2023) [29]. The latter two studies will not be discussed in detail here as the data matrix of Villalobos-Segura et al. (2022) [26] is essentially an extension of that of Villalobos-Segura et al. (2019) [49], while that of Jambura et al. (2023) [29] focuses on sharks rather than batomorphs, and adopted the cod-ing for batomorphs from Villalobos-Segura et al. (2022) [26].

Despite the similar placement of the Late Jurassic batomorph clade, the topology recovered in the present study differs from that of Villalobos-Segura et al. (2022) [26], not least because we have added two newly described species. Furthermore, given the limited knowledge of Late Jurassic ray taxonomy at the time of the study by Villalobos-Segura et al. (2022) [26], several specimens were assigned to what we now know are the wrong species and genera. The present study is therefore the first to analyze specimens that undoubtedly belong to the species †*As. platypterus*, †*B. sismondae*, †*K. etchesi*, and †*S. bugesiacus*, as well as the recently described spe-cies †*Ae. bavarica* (Türtscher et al., 2024 [4]) and the here newly described †*Ap. seioma* gen. et sp. nov. Based on the observations of Türtscher et al. (2024) [4] and the present study, we changed the coding of the Late Jurassic batomorph clade of Villalobos-Segura et al. (2022) [26], on whose data matrix our phylogenetic analysis is based, which also contributed to a changed topology. Below, we explain in detail the most important changes made (see character list for all changes).

Pectoral fin radials (character 83): As the coding of Villalobos-Segura et al. (2022) [26] for †*S. bugesiacus* is mainly based on specimens that were later ascribed to the new species †*Ae. bavarica* (see Türtscher et al., 2024 [4]), we had to change the coding for the character 'pectoral fin radials' (character 114 in Villalobos-Segura et al., 2022 [26]). While in †*Ae. bavarica* (as well as in †*Ap. seioma* gen. et sp. nov., †*As. platypterus*, †*B. sismondae*, and †*K. etchesi*) all radi-als articulate with the three basal cartilages, †*Spathobatis bugesiacus* is the only known Late Jurassic ray in which a radial articulates directly with the scapulocoracoid. We therefore coded this character for †*S. bugesiacus* as 'some articulate directly with scapulocoracoid' (see S5 File).

Postpelvic processes (character 91): The strict consensus tree estimated by Villalobos-Segura et al. (2022) [26] recovered two synapomorphies of Late Jurassic batomorphs, i.e. the presence of an elongated mesopterygium (character 106 in Villalobos-Segura et al., 2022 [26]) and the presence of postpelvic processes (character 118 in Villalobos-Segura et al., 2022 [26]). In light of the new taxonomic diversity of Late Jurassic batomorphs recovered by Türtscher et al. (2024) [4], we re-examined all currently known taxa and agree with Villalobos-Segura et al. (2022) [26] that this group shares the presence of distally projecting mesopterygia that follow the contour of the propterygia. However, we disagree with Villalobos-Segura et al.'s (2022) [26] coding of the postpelvic processes, which are small and shallow rounded processes on the posteromedian margin of the puboischiadic bar (da Silva et al., 2021 [55]). This charac-ter was originally considered a synapomorphy for members of the family Platyrhinidae (Nishida, 1990 [56]; McEachran et al., 1996 [57]; McEachran & Aschliman, 2004 [58]; Aschli-man et al., 2012 [59]), but was later shown to be present in some Torpediniformes and Rhino-pristiformes (da Silva et al., 2021 [55]). According to the study by Villalobos-Segura et al. (2022) [26], postpelvic processes are present in Jurassic batomorphs, Torpediniformes (except *Narke* and *Temera*), Platyrhinidae, Rhinopristiformes (except *Pristis*, *Rhina*, and *Rhynchoba-tus*), and *Hemiscyllium*. According to da Silva et al. (2021) [55], postpelvic processes are absent

in *Aptychotrema*, but we were able to clearly identify them as present in the material examined, in agreement with the interpretation of Villalobos-Segura et al. (2022) [26]. However, Villalobos-Segura et al. (2022) [26] coded this character as present in the Late Jurassic batomorphs †*Asterodermus*, †*Belemnobatis*, †*Kimmerobatis*, and †*Spathobatis*, and recovered the presence of postpelvic processes as a synapomorphy for a Late Jurassic batomorph clade. Our revision of the Late Jurassic batomorphs revealed the presence of the processes only in †*Spathobatis*, while the other Late Jurassic batomorphs, including the recently described †*Aellopobatis* as well as †*Apolithabatis* gen. nov. did not show this character (see S5 File).

Two dorsal fin spines (character 134): Villalobos-Segura et al. (2022) [26] coded dorsal fin spines as absent in †*S. bugesiacus* and as unknown ('?') in †*As. platypterus* (character 130 in Villalobos-Segura et al., 2022 [26]). The supposed †*S. bugesiacus* specimens they examined included three specimens that were previously included in †*S. bugesiacus*, but which we now know to be †*Ae. bavarica* (see Türtscher et al., 2024 [4]) and which do indeed lack dorsal fin spines, and two specimens of †*S. bugesiacus*, one of which lacks the trunk and one of which is a fossil imprint and does not show the minute fin spines. However, re-examination of †*S. bugesiacus* by Türtscher et al. (2024) [4] clearly confirmed that two small fin spines are present in front of each dorsal fin, so we changed the coding of this character to 'present' in †*S. bugesiacus*. Most †*As. platypterus* specimens lack the caudal fin, dorsal fins, and fin spines. In the holotype (NHMUK PV P12067), however, the very small fin spines are preserved, and we therefore have changed the coding for the character 'two dorsal fin spines' to 'present' as well (see S5 File).

Enameloid layer on fin spines (character 135): In accordance with the coding change of character 134 (see above), we also changed the coding of the cohesive character 'enameloid layer on fin spines' to 'present' (character 131 in Villalobos-Segura et al., 2022 [26]) in †*S. bugesiacus* and †*As. platypterus*.

The modified character scoring and the addition of two new taxa resulted in a less resolved tree under the parsimony criterion than in Villalobos-Segura et al. (2022) [26]. While our phylogenetic tree left the interrelationships of Apolithabatiformes ord. nov. largely unresolved, the MP analysis of Villalobos-Segura et al. (2022) [26] suggested a basal position for †*Belemnobatis* and a sister-group relationship between †*Asterodermus* and †*Kimmerobatis*. Note that the maximum likelihood (ML) and Bayesian inference (BI) analyses of Villalobos-Segura et al. (2022) [26] resulted in significantly different topologies, highlighting the uncertainty in the phylogenetic placement of Late Jurassic rays.

Similarly, our maximum likelihood analysis yielded a different result, suggesting that Late Jurassic rays form a paraphyletic group (S6 Fig in S9 File). Although the resulting phylogenetic tree was more resolved than in the MP analysis, these results must be treated with caution; the number of model parameters estimated by W-IQ Tree exceeded the number of characters, and thus phylogenetic estimates should be interpreted with caution. However, both ML and MP analyses agreed on the sister-group relationship between †*Apolithabatis* gen. nov. and †*Aellopobatis*.

These ongoing conflicts in batomorph phylogeny demonstrate that our understanding of the evolution of morphological characters and phylogenetic relationships within batomorphs is not yet fully resolved. There are also serious conflicts between morphological and molecular studies on the relationships between different batomorph orders (e.g., Stein et al., 2018 [60]). Furthermore, the ML and BI analyses of Villalobos-Segura et al. (2022) [26] based on morphological characters show the uncertainty in morphological analyses, with none of the relationships between the different batomorph orders being resolved. A well-supported molecular tree based on whole genomes could provide a backbone for exploring morphological characters in batomorphs, as has been done for sharks (Jambura et al., 2023 [29]). This has the potential to

identify unambiguous synapomorphies, confidently reconstruct the evolution of morphological characters, and thus improve our understanding of the evolutionary history of this group. Our knowledge of batomorph systematics in deep time is still far from complete, as many interrelationships, such as those within apolithabatiform or rhinopristiform taxa, remain largely unresolved. Within Rhinopristiformes, in particular the Late Cretaceous Lebanese batomorphs (i.e., †'*Rhinobatos*' *hakelensis*, †'*Rhinobatos*' *latus*, †'*Rhinobatos*' *maronita*, †'*Rhinobatos*' *whitfieldi*, †*Rhombopterygia*) have been identified as rogue taxa due to their highly uncertain position within the order, resulting in a polytomy. Previous studies have attempted to clarify the phylogenetic relationships of Lebanese batomorphs, but have failed to provide conclusive results (Brito & Dutheil, 2004 [61]; Kachacha et al., 2017 [62]). It is evident that the morphology and taxonomy of these taxa are in urgent need of revision in order to obtain a more resolved phylogenetic tree and thus improve our understanding of the evolution of batomorphs.

## Conclusion

Apart from an as of yet undescribed batomorph skeleton from the Early Jurassic Posidonia Shale near Holzmaden, Germany (Maisey et al., 2020 [3]), which is currently under description, the Late Jurassic taxa †*Ae. bavarica*, †*As. platypterus*, †*B. sismondae*, †*K. etchesi*, and †*S. bugesiacus* represent the earliest known batomorphs preserved as articulated skeletons. In this study we identified another Late Jurassic ray species, †*Apolithabatis seioma* gen. et sp. nov., which is the first batomorph described from Painten, Germany. In addition, this upper Kimmeridgian ray is the oldest Late Jurassic batomorph known from Germany based on skeletal remains. A phylogenetic analysis including †*Ap. seioma* gen. et sp. nov. as well as the recently described †*Ae. bavarica* was conducted. The results suggest that †*Ap. seioma* gen. et sp. nov. and †*Ae. bavarica* are more closely related to each other than to the other members of the newly established stem group batomorph order Apolithabatiformes ord. nov., but also that the interrelationships of Late Jurassic rays are still largely unresolved. The new order should therefore be treated as a working hypothesis that will require further testing after the inclusion of still undescribed specimens. Although the detailed study of these exquisitely preserved fossils has allowed us to make significant progress in understanding the diversity and phylogenetic relationships of early batomorphs, the results also clearly show that we still face crucial obstacles in establishing robust relationships within the batomorph clade based on morphology, and that it is of great importance to put more effort into character exploration in order to obtain a more complete picture of the evolutionary history of rays and skates.

## Supporting information

**S1 File.**
(PDF)

**S2 File. S1-S18 Tables.**
(XLSX)

**S3 File. S1-S3 Figs.**
(PDF)

**S4 File. Traditional morphometrics [% DL] results, S4, S5 Figs, Tables A-G.**
(PDF)

**S5 File. Character list [including explanations and three figures].**
(DOCX)

**S6 File. Data matrix without rogue taxa.**
(NEX)

**S7 File. TNT script.**
(TNT)

**S8 File. Log file.**
(LOG)

**S9 File. S6 Fig.**
(PDF)

**S10 File. Data for morphometric analyses.**
(ZIP)

## Acknowledgments

First of all, we would like to express our sincere thanks to Raimund Albersdörfer. His many years of dedication, his tireless efforts to advance scientific knowledge, and his generosity in providing valuable specimens have made a great contribution to palaeontology in Germany. We are very grateful for the opportunity to study the extraordinary fossil described in this paper. We would also like to thank the staff of the Dinosaurier Museum Altmühltal for their invaluable help in documenting the fossil. Thanks also to Sebastian Stumpf (University of Vienna, Austria) for many fruitful discussions. We are grateful to Jörg Fröbisch for editorial comments and to the three reviewers for their valuable insights and comments on an earlier version of this manuscript.

## Author Contributions

**Conceptualization:** Julia Türtscher, Frederik Spindler.

**Data curation:** Julia Türtscher.

**Formal analysis:** Julia Türtscher, Patrick L. Jambura.

**Funding acquisition:** Jürgen Kriwet.

**Investigation:** Julia Türtscher, Patrick L. Jambura, Frederik Spindler.

**Methodology:** Julia Türtscher, Patrick L. Jambura.

**Project administration:** Julia Türtscher, Jürgen Kriwet.

**Resources:** Jürgen Kriwet.

**Software:** Jürgen Kriwet.

**Supervision:** Jürgen Kriwet.

**Validation:** Julia Türtscher, Patrick L. Jambura, Frederik Spindler, Jürgen Kriwet.

**Visualization:** Julia Türtscher, Patrick L. Jambura, Frederik Spindler.

**Writing – original draft:** Julia Türtscher.

**Writing – review & editing:** Julia Türtscher, Patrick L. Jambura, Frederik Spindler, Jürgen Kriwet.

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
