## [Decision Letter · Decision Letter 0]

9 Jun 2024

PONE-D-24-14855Insights into stem Batomorphii: A new holomorphic ray (Chondrichthyes, Elasmobranchii) from the Upper Jurassic of GermanyPLOS ONE

Dear Dr. Türtscher,

Thank you for submitting your manuscript to PLOS ONE. After careful consideration, we feel that it has merit but does not fully meet PLOS ONE’s publication criteria as it currently stands. Therefore, we invite you to submit a revised version of the manuscript that addresses the points raised during the review process.

**ACADEMIC EDITOR: ** This is a very nice and informative manuscript that provides new insights into stem batomorphs. It is close to being publishable as is, but 2 of the 3 reviewers raised some minor points and suggestions for improvement. Please consider them carefully and address them in your resubmission. I don't have any major objections and only found some very minor typos, which I will list in the following and ask you to correct:

Line 114 correct spelling of 'Kieselplattenkalk'

Line 304 delet 1x "are"

Line 312 correct spelling of describe

Line 549 correct spelling of discernible 

We look forward to receiving your revised manuscript.

Kind regards,

Jörg Fröbisch, Ph.D.

Academic Editor

PLOS ONE

Journal Requirements:

2. Please take this opportunity to be sure you have met all of our guidelines for new species. For proper registration of a new zoological taxon, we require two specific statements to be included in your manuscript.

a.In the Results section, the globally unique identifier (GUID), currently in the form of a Life Science Identifier (LSID), should be listed under the new species name, for example:

Anochetus boltoni Fisher sp. nov. urn:lsid:zoobank.org:act:B6C072CF-1CA6-40C7-8396-534E91EF7FBB

Another LSID for the manuscript itself should also appear within the Nomenclature statement. You will need to contact Zoobank (zoobank.org/About) to obtain a GUID (LSID). You should receive one LSID for your manuscript and a separate, unique LSID for the new species. 

b. Please also insert the following text into the Methods section, in a sub-section to be called ""Nomenclatural Acts"":

The electronic edition of this article conforms to the requirements of the amended International Code of Zoological Nomenclature, and hence the new names contained herein are available under that Code from the electronic edition of this article. This published work and the nomenclatural acts it contains have been registered in ZooBank, the online registration system for the ICZN. The ZooBank LSIDs (Life Science Identifiers) can be resolved and the associated information viewed through any standard web browser by appending the LSID to the prefix ""http://zoobank.org/"". The LSID for this publication is: urn:lsid:zoobank.org:pub: XXXXXXX. The electronic edition of this work was published in a journal with an ISSN, and has been archived and is available from the following digital repositories: PubMed Central, LOCKSS [author to insert any additional repositories].

All PLOS ONE articles are deposited in PubMed Central and LOCKSS. If your institute, or those of your co-authors, has its own repository, we recommend that you also deposit the published online article there and include the name in your article.

Following a recent ruling by the International Commission on Zoological Nomenclature, electronic journals are now a valid format for publication of new zoological taxa. In order to ensure the valid publication of your new species, please be sure to include the updated version of Nomenclatural Acts (above). A complete explanation of our guidelines for publishing new species can be found on our website: http://www.plosone.org/static/guidelines#zoological.

3. Thank you for stating the following in the Acknowledgments Section of your manuscript: "This research was funded in whole by the Austrian Science Fund (FWF) P35357 and P33820 to JK."

Please remove any funding-related text from the manuscript and let us know how you would like to update your Funding Statement. Currently, your Funding Statement reads as follows: "This research was funded in whole by the Austrian Science Fund (FWF) P35357 and P33820 to JK. The funders had no role in study design, data collection and analysis, decision to publish, or preparation of the manuscript.

https://www.fwf.ac.at/en/"

Additional Editor Comments:

This is a very nice and informative manuscript that provides new insights into stem batomorphs. It is close to being publishable as is, but 2 of the 3 reviewers raised some minor points and suggestions for improvement. Please consider them carefully and address them in your resubmission. I don't have any major objections and only found some very minor typos, which I will list in the following and ask you to correct:

Line 114 correct spelling of 'Kieselplattenkalk'

Line 304 delet 1x "are"

Line 312 correct spelling of describe

Line 549 correct spelling of discernible

Reviewers' comments:

Reviewer's Responses to Questions

**Comments to the Author**

1. Is the manuscript technically sound, and do the data support the conclusions?

Reviewer #1: Yes

Reviewer #2: Yes

Reviewer #3: Yes

2. Has the statistical analysis been performed appropriately and rigorously? 

Reviewer #1: Yes

Reviewer #2: Yes

Reviewer #3: Yes

3. Have the authors made all data underlying the findings in their manuscript fully available?

Reviewer #1: Yes

Reviewer #2: Yes

Reviewer #3: Yes

4. Is the manuscript presented in an intelligible fashion and written in standard English?

Reviewer #1: Yes

Reviewer #2: Yes

Reviewer #3: Yes

5. Review Comments to the Author

Reviewer #1: The manuscript is an interesting paper that contributes to the knowledge of the paleoichthyology of the Late Jurassic of Germany. The manuscript is clear, concise and well written. Figures are of high quality. Results and DIscussion are supported by data. For these reason I suggest to accept the manuscript as it is.

Reviewer #2: Dear Editor and Authors,

The study reports of a new batomorph species from the Painten locality (Solnhofen Arcipelago, Upper Jurassic). Data are presented in an appropriate fashion and statistics are more than adequate. The inferences are well-supported and the language is fluent. All in all, this study is of international and broad interest and provides novel and vey significant information.

There are only some minor points I would like to raise:

- I would recommend inserting a figure for the geological/paleogeographic/geographic setting to better display all the localities of the Solnhofen Arcipelago and the relationships with Painten. It would be also reasonable to include a stratigraphic scheme of the quarry with indication of the level of provenance of the new taxon.

- There are minor typos in the manuscript, please check all the issues throughout the whole manuscript, I might have missed some of them (indications in the annotated reviewer attachment)

- Landmarks are described in a caption, I would recommend describing them in a paragraph in the method section or in supplementaries and shortening the caption.

- Please consider inserting a paragraph with salient diagnostic characters of the new order as indicated in the annotated attachment, just as a brief summary of the character discussed.

- line 578-582: please consider dividing the sentences, it is quite complex.

- The discussion section does not include any hint of the stratigraphic distribution of the new taxon in comparison with the other identified taxa, while in the conclusion paragraph it is said that the new taxon is the oldest Jurassic batomorph. I would recommend emphasizing and discussing extensively this in the discussion paragraph, I think it could be significant. Is there any pattern in the stratigraphic distribution of the mentioned taxa? Also a figure could be useful for the comparison.

- The phylogenetic hypothesis is very interesting and the authors also stressed it to be treated with caution, but I think that the result is significant. I would recommend emphasizing also in the conclusion paragraph that the new order should be treated as a working hypothesis, unresolved for still undescribed material from other Lagerstatten; it might be redundant but I think it is always good.

I would be pleased to answer any question from the authors and be open to any discussion.

Best regards,

Jacopo Amalfitano, PhD

Reviewer #3: This is an excellent piece of work further emphasising the diversity of Late Jurassic batoids. The descriptions, discussions and conclusions are all good.

I would suggest looking at the ordering of sections- the large sections on landmarks and other methods comes before anything on the fossil and therefore suggests considerable prior knowledge of batoid anatomy.

The description is good, but the section on teeth and denticles is very brief- I know teeth are not clear but presumably there is a large denticle coverage and it is this that gives the outline.

The phylogeny looks good and makes a lot of sense.

6. PLOS authors have the option to publish the peer review history of their article (what does this mean?). If published, this will include your full peer review and any attached files.

Reviewer #1: **Yes: **Giuseppe Marramà

Reviewer #2: **Yes: **Jacopo Amalfitano

Reviewer #3: **Yes: **Charlie Underwood

---

## [Author Response · Author response to Decision Letter 0]

14 Aug 2024

Response to Reviewers 

Editor Comments: This is a very nice and informative manuscript that provides new insights into stem batomorphs. It is close to being publishable as is, but 2 of the 3 reviewers raised some minor points and suggestions for improvement. Please consider them carefully and address them in your resubmission. 

Response: Thank you very much! We have addressed all of the reviewers' comments (see revised document and comments below) and are confident that this revision has improved the quality of the manuscript and brought it to a publishable state. 

I don't have any major objections and only found some very minor typos, which I will list in the following and ask you to correct: 

 Line 114 correct spelling of 'Kieselplattenkalk' 

 Line 304 delete 1x "are" 

 Line 312 correct spelling of describe 

 Line 549 correct spelling of discernible 

Response: Thank you for pointing out all these typos; we corrected them. 

Reviewer #1: The manuscript is an interesting paper that contributes to the knowledge of the palaeoichthyology of the Late Jurassic of Germany. The manuscript is clear, concise and well written. Figures are of high quality. Results and Discussion are supported by data. For these reasons I suggest to accept the manuscript as it is. 

Response: Thank you very much for reviewing the manuscript, we are delighted to read these positive comments! 

Reviewer #2: Dear Editor and Authors, the study reports of a new batomorph species from the Painten locality (Solnhofen Archipelago, Upper Jurassic). Data are presented in an appropriate fashion and statistics are more than adequate. The inferences are well-supported and the language is fluent. All in all, this study is of international and broad interest and provides novel and very significant information. 

Response: Thank you for reviewing the manuscript and for your positive and constructive feedback! 

There are only some minor points I would like to raise: 

 I would recommend inserting a figure for the geological/paleogeographic/geographic setting to better display all the localities of the Solnhofen Archipelago and the relationships with Painten. It would be also reasonable to include a stratigraphic scheme of the quarry with indication of the level of provenance of the new taxon. 

Response: We agree and thus included a figure with the geographical setting as well as the stratigraphy of Painten. For the palaeogeographic setting, we refer in the text to Villalobos-Segura et al. (2023): A synoptic review of the cartilaginous fishes (Chondrichthyes: Holocephali, Elasmobranchii) from the Upper Jurassic Konservat-Lagerstätten of southern Germany: taxonomy, diversity, and faunal relationships, as this very recent publication includes a geographical and palaeogeographical map of the Solnhofen Archipelago with biostratigraphical information. 

 There are minor typos in the manuscript, please check all the issues throughout the whole manuscript, I might have missed some of them (indications in the annotated reviewer attachment). 

Response: Thank you very much for pointing out these typos, we checked the manuscript and corrected them. 

 Landmarks are described in a caption; I would recommend describing them in a paragraph in the method section or in supplementaries and shortening the caption. 

Response: We agree that the figure caption was too long with the landmark descriptions included. Therefore, we moved this part to the respective Method-sections. 

 Please consider inserting a paragraph with salient diagnostic characters of the new order as indicated in the annotated attachment, just as a brief summary of the character discussed. 

Response: As suggested, we included a paragraph with the diagnostic characters of the new order. 

 line 578-582: please consider dividing the sentences, it is quite complex. 

Response: As suggested, we divided this complex sentence. Previously this sentence read “The dorsal fins appear to have been flipped over first, followed by the rigid spine because of taphonomic processes, resulting in the supraneural spines piercing the already decomposing skin and the flipped dorsal fins becoming 'stuck' between supraneural spines and the folded skin on the right side of the tail.”. The split, new sentence now reads: “Due to taphonomic processes, the dorsal fins appear to have been flipped over first, followed by the rigid spine. This apparently resulted in the supraneural spines piercing the already decomposing skin, and the flipped dorsal fins becoming 'stuck' between the supraneural spines and the folded skin on the right side of the tail.” 

 The discussion section does not include any hint of the stratigraphic distribution of the new taxon in comparison with the other identified taxa, while in the conclusion paragraph it is said that the new taxon is the oldest Jurassic batomorph. I would recommend emphasizing and discussing extensively this in the discussion paragraph, I think it could be significant. Is there any pattern in the stratigraphic distribution of the mentioned taxa? Also, a figure could be useful for the comparison. 

Response: Thank you for this suggestion. We revised the first paragraph of the discussion, added stratigraphic information about the taxa, and emphasised that the new taxon is the oldest Jurassic batomorph from southern Germany. 

 The phylogenetic hypothesis is very interesting and the authors also stressed it to be treated with caution, but I think that the result is significant. I would recommend emphasizing also in the conclusion paragraph that the new order should be treated as a working hypothesis, unresolved for still undescribed material from other Lagerstätten; it might be redundant but I think it is always good. 

Response: We agree and have included this statement in the conclusions as follows: “The results suggest that †Ap. seioma gen. et sp. nov. and †Ae. bavarica are more closely related to each other than to the other members of the newly established stem group batomorph order Apolithabatiformes ord. nov., but also that the interrelationships of Late Jurassic rays are still largely unresolved. The new order should therefore be treated as a working hypothesis that will require further testing after the inclusion of still undescribed specimens.”. 

Reviewer #3: This is an excellent piece of work further emphasising the diversity of Late Jurassic batoids. The descriptions, discussions and conclusions are all good. 

Response: Thank you for reviewing the manuscript and for your positive and constructive feedback! 

 I would suggest looking at the ordering of sections - the large sections on landmarks and other methods comes before anything on the fossil and therefore suggests considerable prior knowledge of batoid anatomy. 

Response: We see your point, but we decided to keep the current arrangement for three reasons: (1) The "Materials and Methods" section is in the same order as the "Results" section (i.e., Anatomy, Traditional Morphometrics, Geometric Morphometrics, Phylogeny), which we think makes sense. (2) It would be possible to include the Systematic Palaeontology below "Materials" and before the rest of the "Materials and Methods" section, but we feel that the description of the specimen belongs more to the results than to MM, and that this arrangement would split up the "Materials and Methods", potentially causing more confusion than the current arrangement. (3) We have included several figures and line drawings of the specimen with the anatomical characters labelled, as well as illustrations showing the measurements taken and the landmark configuration, and are therefore confident that readers without in-depth anatomical knowledge of batomorphs will still be able to follow our methods and results. Thank you very much for this constructive comment, and we hope that you will agree with us leaving the structure as it is. 

 The description is good, but the section on teeth and denticles is very brief - I know teeth are not clear but presumably there is a large denticle coverage and it is this that gives the outline. 

Response: We agree and have included a short paragraph as well as a figure on the denticles in the description. However, it is not possible to go into more detail on tooth and denticle morphology at this time, as we do not have microscopic images and cannot provide them at this time. As the focus of this paper lies on skeletal anatomy and body outline, we also feel that a more detailed description of teeth and denticles is not necessary at this point. Finally, we would like to add that there are plans for future projects on holomorphic Late Jurassic batomorphs which will focus on tooth and denticle morphology and will include the specimen described in this paper. 

 The phylogeny looks good and makes a lot of sense. 

Response: Thank you very much!

---

## [Editor Report · Decision Letter 1]

27 Aug 2024

Insights into stem Batomorphii: A new holomorphic ray (Chondrichthyes, Elasmobranchii) from the Upper Jurassic of Germany

PONE-D-24-14855R1

Dear Dr. Türtscher,

We’re pleased to inform you that your manuscript has been judged scientifically suitable for publication and will be formally accepted for publication once it meets all outstanding technical requirements.

Kind regards,

Jörg Fröbisch, Ph.D.

Academic Editor

PLOS ONE

---

## [Editor Report · Acceptance letter]

20 Sep 2024

PONE-D-24-14855R1 

PLOS ONE

Dear Dr. Türtscher, 

I'm pleased to inform you that your manuscript has been deemed suitable for publication in PLOS ONE. Congratulations! Your manuscript is now being handed over to our production team.

Kind regards, 

on behalf of

Prof. Jörg Fröbisch 

Academic Editor

PLOS ONE